# LILO: Learning to Reason at the Frontier of Learnability

**Thomas Foster**[†][*]  **Anya Sims**[†]

**Mattie Fellows**[†]  **Johannes Forkel**[†]  **Jakob Foerster**[†]

## Abstract

Reinforcement learning is a widely adopted component of large language model post-training, especially for reasoning-style tasks such as maths questions. However, as we show, most existing methods will provably fail to learn from questions that are too hard, where the model always fails, or too easy, where the model always succeeds. Much human effort is therefore spent producing datasets of questions of a suitable difficulty for state-of-the-art models. Given this, we consider how to algorithmically identify questions that allow for maximally efficient training. We introduce a method, LILO (*Learnability Improves LLMs Optimally*), that prioritises training on questions with high variance of success, known as *learnability*, and we provide theory which shows that LILO enables the expected improvement of the model to be large. We run a wide range of experiments over multiple base models, algorithms and reasoning datasets to demonstrate that LILO consistently reaches a higher final test accuracy, and can do so in $3\times$ fewer training steps. We explore how questions with high learnability can be efficiently identified, and discuss how learnability can be scaled to produce LLM agents that autonomously and open-endedly expand the frontier of human knowledge.

## 1 Introduction

Reinforcement learning (RL) has become a crucial post-training step of many state-of-the-art reasoning-focused large language models (LLMs), notably DeepSeek-R1 [1], Tulu [2], and OpenAI's O1 [3]. Policy-gradient algorithms—such as PPO, VinePPO, and GRPO—have become the de facto standard, using *advantage estimation* to reinforce promising sections of answers and penalise errors. However, as we show, many existing methods will provably fail to learn from questions that are too hard, where the model always fails, or too easy, where the model always succeeds. Given that RL with LLMs is extremely compute intensive, training on such questions is a huge waste of resources. Much human effort and millions of pounds are therefore spent continually producing new datasets of suitable difficulty for state-of-the-art models to train on.

In this work, we ask a fundamental question: **Can we algorithmically identify questions that are optimal to learn from?** To answer this, we revisit the formal framing of teaching LLMs to reason with RL, connecting it to the existing literature of automatic curriculum learning (ACL), unsupervised environment design (UED) and *learnability*—**defined as the variance of success of a model over multiple attempts at a given question.** We fill a vital gap in the literature, with a proof that the expected improvement of the model is constrained by the the learnability of the questions attempted during training.

This motivates us to develop a novel method for prioritising training on questions with high learnability. **We name it LILO- "*Learnability Improves LLMs Optimally*"**. LILO continuously adapts to the model's ability during training, identifying questions that the model can answer correctly, but

39th Conference on Neural Information Processing Systems (NeurIPS 2025).

not consistently, and which, as we will show, are theoretically most useful to learn from. The implementation of LILO is extremely simple, and we demonstrate it can be added seamlessly to existing training libraries (OatLLM [4] and VinePPO [5]) **in fewer than 20 lines of code**. Running LILO requires minimal extra compute cost, exploiting the current trend of generating many attempts per question when teaching LLMs to reason with RL.

We evaluate LILO across three RL algorithms (GRPO [6], PPO [7] and VinePPO [5]), three training datasets of varying size and difficulty (the often standard GSM8K [8] dataset, the more challenging MATH [9] dataset, and the larger and more diverse ORZ57K [10]) and two base models (Rho-1B [11] and Qwen-2.5-1.5B±[12]). In doing so, we are the first to present evidence that existing training methods waste significant compute with training on questions with zero learnability. In contrast, LILO prioritises training on high learnability questions which **increases final test accuracy by several percentage points, and can do so in** $3\times$ **fewer training steps**. Furthermore, we evaluate LILO on a variety of unseen test datasets, such as CollegeMath [13] and OlympiadBench [14], and find that adding LILO improves reasoning on out-of-distribution questions.

Our theory and results show that Learnability provides a principled approach to selecting training data for reasoning models, and we conclude with a discussion of future work required to develop this line of research into fully open-ended curricula for language model agents.

To summarise our contributions:

- **Theory:** Section 3 proves Theorem 3.1, which provides a mathematical argument that expected policy improvement is upper bounded by a quantity which increase linearly with learnability.
- **Method:** Section 4 presents a simple and efficient algorithm for prioritising training on questions with high learnability.
- **Results:** Section 6 presents the first results that prioritising learnability during the RL training of LLMs improves training speed by $3\times$ whilst boosting final performance.
- **Insights:** Section 7 contributes numerous valuable insights for training LLMs with RL, uncovering 1) how questions selected by LILO correlate with human-interpretable factors, 2) that learnable questions become progressively harder to find and 3) how even training on highly learnable questions doesn't always boost performance, due in part to reasoning models' large train–test generalisation gap

## 2 Background

### 2.1 Training LLMs to reason with RL

The goal of training LLMs with RL is to maximise the expected reward $J(\theta) = \mathbb{E}_{\mathbf{q}\sim\mathcal{P},\ \mathbf{a}\sim\pi_\theta}[r(\mathbf{q}, \mathbf{a})]$, where $\mathbf{q}$ is a question sampled from a distribution $\mathcal{P}$, and $\mathbf{a}$ is an answer to the question generated by LLM $\pi_\theta$. To find the parameters $\theta^* \in \Theta$ which maximise $J(\theta)$, answer generation is framed as a Markov Decision Process (MDP), where each state $s_t$ in the MDP represents a sequence of tokens, and a policy autoregressively chooses the next token $a_t \sim \pi_\theta(a_t|s_t)$ to be added to the current sequence. For reasoning problems, only once the entire answer has been generated is a reward received, i.e for a generation of length $T$, for $t = 0, ..., T - 1$ we have $r(s_t) = 0$, and only $r(s_T)$ can be non-zero. This MDP permits several other equivalent formulations that closely connect it to existing work on ACL, UED and learnability. We discuss this further in Section 8 and Appendix G.

Policy gradient algorithms, such as RLOO [15], GRPO [6], PPO [7] and VinePPO [5] are the de-facto standard for training LLMs to reason with RL, optimising $J(\theta)$ via the policy gradient theorem $\nabla_\theta J(\theta) = \mathbb{E}_{\pi_\theta}\left[\sum_{t=0}^{T-1} \nabla_\theta \log \pi_\theta(a_t|s_t) \cdot A_t\right]$. The advantages $A_t$, $t = 0, ..., T - 1$, characterise how much better a given choice of token $a_t$ is compared to the model's current behaviour. The exact way the advantages are computed, and how $J(\theta)$ is optimised for, varies for different policy gradient algorithms (see Appendix A for more details). In Section 3 we analyse how these choices affect the relationship between expected policy improvement and learnability.

### 2.2 Learnability

Learnability is a simple way to asses the difficulty of a question with a binary outcome. It is defined in [16, 17, 18] as **Learnability**$(\pi_\theta) := \text{Var}_{\pi_\theta}[r(s_T)] = \mathbb{E}_{\pi_\theta}\left[\left(r(s_T) - \mathbb{E}_{\pi_\theta}[r(s_T)]\right)^2\right] =$

$p_\theta(1 - p_\theta)$ where $p_\theta = \mathbb{E}_{\pi_\theta}[r(s_T)]$. Given $K$ attempts at a question, we estimate learnability by computing the empirical success rate $\hat{p} = \#\text{successes}/K$, with learnability then being $\hat{p}(1 - \hat{p})$, or $\frac{K}{K-1}\hat{p}(1 - \hat{p})$ with Bessel correction.

For many algorithms, it is simple to show the outcome of training on questions with zero learnability. For example, consider RLOO and GRPO, that compute advantage as $A_t = r(s_T) - \mathbb{E}_{\pi_\theta}[r(s_T)]$ or $A_t = \frac{r(s_T) - \mathbb{E}_{\pi_\theta}[r(s_T)]}{\text{Var}_{\pi_\theta}[r(s_T)]}$ where the success probability $\mathbb{E}_{\pi_\theta}[r(s_T)]$ and success variance $\text{Var}_{\pi_\theta}[r(s_T)]$ are estimated from a batch of attempts sampled from the model. When learnability is zero, it holds that $A_t = 0$ for all $t = 0, ..., T - 1$, and this question has no contribution to the model update.

There is minimal prior work examining the impact of training on questions with learnability $> 0$. Whilst, the authors of [16] show the expected improvement of a model trained by the classic algorithm REINFORCE [19] is maximised by maximising learnability, they require a very simplified learning setting with strong assumptions.

## 3    Relationship between Expected Policy Improvement and Learnability

In this section we introduce Theorem 3.1, which provides a mathematical argument that the policy improvement resulting from one gradient update is upper bounded by a quantity which increases linearly with learnability, when using RLOO, GRPO, VinePPO, PPO or any other advantage based policy gradient algorithm. A full proof of Theorem 3.1 is provided in Appendix B.

We first see that when setting $\theta_{k+1} = \theta_k + \beta\nabla_\theta J(\theta_k)$, for a learning rate $\beta > 0$, and assuming that $\|\theta_{k+1} - \theta_k\| = \beta\|\nabla_\theta J(\theta_k)\|$ is small, we can use the first-order Taylor expansion of $J(\theta_{k+1}) = J(\theta_k + \beta\nabla_{\theta_k}J(\theta_k)) \approx J(\theta_k) + \beta\|\nabla_{\theta_k}J(\theta_k)\|^2$. This yields that

$$J(\theta_{k+1}) - J(\theta_k) \approx \beta\|\nabla_{\theta_k}J(\theta_k)\|^2, \tag{1}$$

i.e. the policy improvement is proportional to the gradient magnitude. Theorem 3.1 then connects the gradient magnitude to learnability:

---

**Theorem 3.1**

Consider an MDP in which reward is obtained only in the terminal state $s_T$, and a parametrised policy $\pi_\theta$. Assume that $\left(\nabla_\theta \log \pi_\theta(a_t|s_t)\right)^T \nabla_\theta \log \pi_\theta(a_{t'}|s_{t'})$ and $A_t A_{t'}$ are uncorrelated for all $t, t' = 0, ..., T - 1$. Then the following statements hold:

For $A_t = r(s_T)$, i.e. REINFORCE:

$$\|\nabla_\theta J(\theta)\|^2 \le \mathbb{E}_{\pi_\theta}\left[\|\sum_{t=0}^{T-1}\nabla_\theta\log\pi_\theta(a_t|s_t)\|^2\right]\mathbb{E}_{\pi_\theta}\left[r(s_T)^2\right].$$

For $A_t = r(s_T) - \mathbb{E}_{\pi_\theta}[r(s_T)]$, i.e. RLOO:

$$\|\nabla_\theta J(\theta)\|^2 \le \mathbb{E}_{\pi_\theta}\left[\left\|\sum_{t=0}^{T-1}\nabla_\theta\log\pi_\theta(a_t|s_t)\right\|^2\right]\mathbb{E}_{\pi_\theta}\left[\left(r(s_T) - \mathbb{E}_{\pi_\theta}\left[r(s_T)\right]\right)^2\right].$$

For $A_t = V^{\pi_\theta}(s_{t+1}) - V^{\pi_\theta}(s_t)$, i.e. PPO, VinePPO:

$$\|\nabla_\theta J(\theta)\|^2 \le \sum_{t=0}^{T-1}\mathbb{E}_{\pi_\theta}\left[\|\nabla_\theta\log\pi_\theta(a_t|s_t)\|^2\right]\mathbb{E}_{\pi_\theta}\left[A_t^2\right]$$

$$= \mathbb{E}_{\pi_\theta}\left[\|\nabla_\theta\log\pi_\theta(a_0|s_0)\|^2\right]\mathbb{E}_{\pi_\theta}\left[\left(r(s_T) - \mathbb{E}_{\pi_\theta}\left[r(s_T)\right]\right)^2\right],$$

where the second equality holds under the additional assumption that $\mathbb{E}_{\pi_\theta}\left[\|\nabla_\theta\log\pi_\theta(a_t|s_t)\|^2\right]$, $t = 0, ..., T - 1$, are all equal. Even without this additional assumption, it still holds that $\sum_{t=0}^{T-1}\mathbb{E}_{\pi_\theta}\left[A_t^2\right] = \mathbb{E}_{\pi_\theta}\left[\left(r(s_T) - \mathbb{E}_{\pi_\theta}\left[r(s_T)\right]\right)^2\right]$.

---

**Theorem 3.1, combined with Equation 1, provides a mathematical argument that the policy improvement obtained from one gradient update is upper bounded by the variance of the final reward, times an additional factor.** When using an algorithm which follows the gradient of a clipped objective, one can expect that a slightly modified version of this argument still applies, since all the learning of such algorithms occurs when in the unclipped portion of the objective.

In the case of binary rewards i.e. $r(s_T) \in \{0, 1\}$, such as in reasoning questions, it holds that $\mathbb{E}_{\pi_\theta}\left[\left(r(s_T) - \mathbb{E}_{\pi_\theta}[r(s_T)]\right)^2\right] = p_\theta(1 - p_\theta)$, for $p_\theta = \mathbb{E}_{\pi_\theta}[r(s_T)]$. Thus for **RLOO**, **PPO** and **VinePPO**, assuming approximately correct estimates of $V^{\pi_\theta}$ through multiple rollouts (VinePPO) or a learned value network (PPO), Theorem 3.1 suggests that the magnitude of the policy gradient, and thus by Equation 1 the expected policy improvement, is upper bounded by learnability times an additional factor.

**GRPO** uses a normalised advantage function, i.e. $A_t = (r(s_T) - \mathbb{E}_{\pi_\theta}[r(s_T)])/\sqrt{\mathrm{Var}_{\pi_\theta}[r(s_T)]}$. Thus we see that under the assumptions of Theorem 3.1 it holds that $\mathbb{E}_{\pi_\theta}\left[\|\nabla_\theta J(\theta)\|^2\right] = \mathbb{E}_{\pi_\theta}\left[\|\sum_{t=0}^{T-1} g_{t,\theta}\|^2\right]$. In other words, the normalisation of the advantage function removes the dependency of the expected gradient magnitude on learnability. However, multiple attempts to replicate Deepseek R1-Zero [1] have found much better performance without this normalisation, i.e. when using $A_t = r(s_T) - \mathbb{E}_{\pi_\theta}[r(s_T)]$, as in RLOO, with which the expected policy improvement is again upper bounded by a quantity which increases linearly with learnability.

**Significance tests:** We performed numerous statistical tests to validate our assumptions and test Theorem 3.1 end to end. These are detailed in Appendix C.

## 4 Method

In Section 3, we argued that the expected policy improvement is small when training on questions with smalllearnability. Thus, in Algorithm 1 we present a method that, at every training step 1) produces a batch of questions with high learnability and 2) trains on this batch. In Algorithm 2, we introduce a simple method to produce a batch of questions with high learnability, based on rejection sampling. The idea is to first estimate the learnability of a pool of $|D|$ candidate questions using a small number $N_{\text{learnability}}$ of attempts per question. A larger number $N_{\text{train}}$ of attempts per question is then rolled out during training on the top-$|B|$ learnable questions.

---

**Algorithm 1** Training with LILO

**Input:** Initial model parameters: $\theta$, Number train steps: $T$, Training batch size: $|B|$

1: **for** $t = 1$ **to** $T$ **do**
2:     $B \leftarrow$ **get_learnable_questions**$(\theta, |B|)$ e.g using Algorithm 2
3:     $\theta \leftarrow$ **train_on_batch**$(\theta, B)$
4: **end for**

---

**Algorithm 2** Get Learnable Questions by Rejection Sampling

**Input:** Model parameters: $\theta$, Size of question batch to return: $|B|$, Size of candidate pool: $|D|$, Number of attempts per question to calculate learnability: $N_{\text{learnability}}$

1: Sample candidate pool of $|D|$ questions from dataset
2: Rollout $N_{\text{learnability}}$ attempts per candidate question
3: Compute success rate per question = $\hat{p}$
4: Compute learnability per question = $\hat{p}(1 - \hat{p})$
5: Return top-$|B|$ questions by learnability

---

Many large-scale implementations of RLOO, GRPO, PPO and VinePPO for teaching LLMs to reason with RL use large values of $N_{\text{train}}$, e.g VinePPO [5] uses $N_{\text{train}} \approx 500$. In this case, taking $N_{\text{learnability}} = 8$ only requires $4\%$ more samples, which is a negligible increase in wall-clock time. The smaller scale implementations of PPO and GRPO that we use in some experiments take $N_{\text{train}} = 8$, in which case we simply reuse the $N_{\text{learnability}}$ samples from the selected top-$|B|$ questions generated

during line 2 of Algorithm 1, and require no further samples during line 3. In this case, the sampling overhead is approximately $4\times$, which we do not consider an issue for the following reasons:

1. For larger scale implementations with larger values of $N_{\text{train}}$, rejection sampling adds minimal overhead.

2. Many of our results show that learnability prioritised training plateaus at a higher level, and so even allowing baselines infinitely more samples would not give the same level of final performance.

3. In this work, we are interested in studying whether training on maximally learnable batches can improve performance in LLMs, and we therefore leave developing algorithms to compute learnability more optimally to future work. In Section 9, we provide a discussion of potential directions towards more sample efficient learnability-based algorithms, including an algorithm that finds highly learnable questions with **no extra sampling overhead** in Appendix D.1 for use in situations where sampling speed is a bottleneck.

4. As we summarise in Table 1, using learnability to generate samples and then discarding some is a more efficient way to scale than training on all $4\times$ samples, as measured by wall-clock time. This is because generation can be done with highly specialised inference engines on independent distributed nodes. Increasing training throughput is much harder, requiring the communication of weights and gradients between nodes. For this reason, leveraging additional samples during training LLMs with RL is an increasingly popular trend to compute value functions by VinePPO [5], and to rapidly speed up training with AsyncRLHF [20] and OpenInstruct [21]. Using reward models with synthetic data also fits this trend, requiring thousands of samples to fit the reward model before RL training commences.

| Paradigm | Algorithm | Sampling time(s) | Training time(s) | Total(s) |
|---|---|---|---|---|
| $N_{\text{train}} \approx N_{\text{learnability}}$ | **PPO** | 20 | 200 | 220 |
| i.e. Small $N_{\text{train}}$ PPO, GRPO | **PPO + LILO** | 80 | 200 | 280 |
| $N_{\text{train}} >> N_{\text{learnability}}$ | **VinePPO** | 1500 | 500 | 2000 |
| i.e. Large $N_{\text{train}}$ PPO, GRPO, VinePPO | **VinePPO + LILO** | 1550 | 500 | 2050 |

Table 1: Prioritising learnability with Algorithm 2 has minimal impact on the total time for a single training iteration. Numbers taken from PPO and VinePPO runs shown in Section 6. Appendix D.1 discusses how high learnability batches can be computed without additional samples, for scenarios where sampling speed is a bottleneck.

## 5 Experimental setup

**RL algorithms:** We add LILO to two existing open-source libraries for training LLMs to reason with RL. The VinePPO library [5] provides implementations of PPO and VinePPO. The OAT library [4] implements GRPO tuned to closely replicate the Deepseek R1 results. All of the hyperparameters for PPO, VinePPO and GRPO we leave unchanged from their implementations in [5] and [4]. We choose these algorithms to cover two major axes of variation for policy gradient algorithms: 1) the coarseness of credit assignment (GRPO assigns credit at the sequence level, PPO/VinePPO and token level), 2) the use of value functions (VinePPO does empirical rollouts, PPO uses value networks). All our algorithms follow the clipped objective, which has become standard.

**Base models:** For our VinePPO and OAT experiments, we follow their baseline setup, using Rho-Math-1B[11] and Qwen-2.5-1.5B [12] respectively. In Section 7, we additionally explore whether LILO can be used to squeeze further performance out of models that have already undergone heavy RL training to reach state-of-the-art, and so we experiment with Oat-Zero-1.5B [22] as the base model.

**Hyperparameters:** The main hyperparameters for rejection sampling in Algorithm 2 is the size of candidate pool $|D|$, and the value of $N_{\text{learnability}}$, the number of responses sampled per question. Ideally, these would be tuned to be as small as possible whilst achieving batches with high learnability. In practice, choosing $|D| = 4 \times |B|$ and $N_{\text{learnability}} = 8$ works well with minimal overhead. The only exception is for VinePPO on GSM8K, where the model nearing 95% train accuracy requires $|D| = 8 \times |B|$ to produce high learnability batches.

| Algorithm | Train Dataset | Speed-up (Steps) | Final test accuracy (%) |
|---|---|---|---|
| PPO | MATH | 2.5x | $19.1 \rightarrow 21.8$ |
| | GSM8K | 1.9x | $51.1 \rightarrow 53.2$ |
| VinePPO | MATH | 3.2x | $22.8 \rightarrow 24.9$ |
| | GSM8K | 3.3x | $53.2 \rightarrow 55.9$ |
| GRPO | ORZ57K | 1.5x | $35.5 \rightarrow 37.1$ |

Table 2: Summary of results. We compute speed-up by comparing the training step at which the highest test accuracy of the baseline is reached by LILO. Final test accuracy compares the performance of the baseline algorithm to the final test accuracy achieved by the algorithm+LILO.

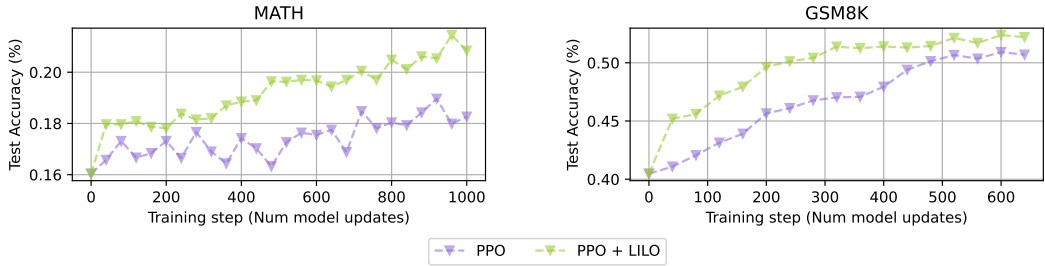

Figure 1: Adding LILO to PPO increases both the model improvement rate and final test accuracy when training on MATH and GSM8K.

**Datasets:** For VinePPO and PPO experiments, we train on mathematical reasoning datasets MATH [9], ( 12,000 competition-level problems), and GSM8K [8], (8,000 simpler grade school problems). These are chosen to demonstrate the effect of learnability prioritised training in situations where initial model performance is high (GSM8K) and where it is initially low (MATH). We further evaluate downstream performance of the MATH-trained models on CollegeMATH [13] (2,818 college-level questions) and OlympiadBench [14] (8,000 Olympiad level maths and physics competitions). For the GRPO experiments, we use the OAT-library and train on the ORZ57K dataset [10] of 57,000 questions amalgamated from AIME [23], Numina-Math [23], Tulu3 MATH [2] and others. For evaluation, we follow their setup and test on 5 datasets MATH [9], Minerva [24], Olympiad Bench [14], AMC [23] and AIME [23].

**Metrics:** We evaluate model performance on the test sets of each dataset, using accuracy (Pass@1) as the primary metric. For PPO/VinePPO this is computed using extract string matching as per the VinePPO library. For GRPO, the OAT library is more flexible, attempting to match the answer to many other equivalent mathematical forms.

## 6 Results

**Adding LILO to PPO:** We train on MATH and GSM8k using the VinePPO codebase. For MATH, we follow the baseline and run for 1000 timesteps (approximately 8 epochs of the data). The improvement rate is almost doubled by adding LILO, increasing by 4.8% compared to 2.1% over the course of training. PPO+LILO reaches the best performance achieved by PPO in 2.5x fewer training steps. When training on GSM8K the runs plateau after 650 steps, with PPO+LILO achieving a slightly higher final test accuracy. It achieves this final test accuracy in 1.9x fewer steps than PPO without Learnability.

**Adding LILO to VinePPO** We again compare on MATH and GSM8K and find that VinePPO also benefits significantly from prioritising learnability. On MATH, it achieves 2.1% higher best test accuracy, and achieves the best test accuracy of VinePPO without learnability in 3.2x fewer steps. On GSM8K we find that finding learning questions via rejection sampling (Algorithm 2) with $|D| = 4 \times |B|$ cannot consistently learnable questions during training, as train performance is closing in on 95%. We increase $|D| = 8 \times |B|$, which is enough to find learnable questions consistently enough during training to boost performance. VinePPO without learnability improves performance on GSM8K from 40.5% to 53.2%, a gain of 12.7%. Prioritising learnability improves on this, taking

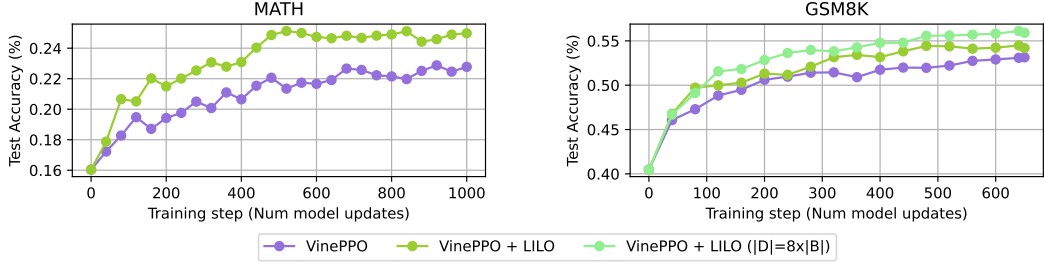

Figure 2: Adding LILO to VinePPO increases both the model improvement rate and final test accuracy when training on MATH and GSM8K. During training on GSM8K, train accuracy reached nearly 95% (see Figure 9), and LILO began to struggle to find high learnability questions (as shown in Figure 6). Doubling the size of the candidate pool $|D|$ in Algorithm 2 fixed this and further improved performance (mint green line).

test accuracy from 40.5% to 55.9%, a gain of 15.4%. VinePPO with learnability reaches the best accuracy achieved without learnability in 3.3x fewer training steps.

We evaluate the four combinations of PPO/VinePPO training on MATH/GSM8K on the holdout datasets CollegeMath and Olympiad Bench. On all four combinations, learnability improves performance, for full results see Table 6 in Appendix F.

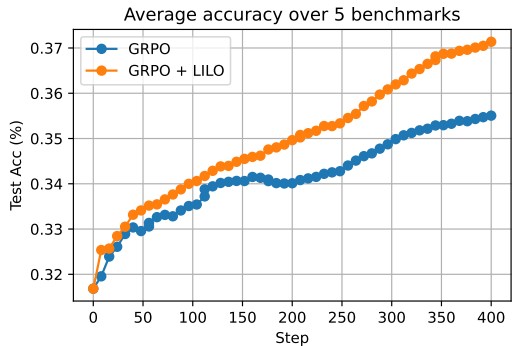

Figure 3: Adding LILO to GRPO training on ORZ57K nearly doubles the model improvement rate.

**Adding LILO to GRPO** Following the OAT library, we train Qwen-2.5-1.5B on ORZ57K using GRPO. Learnability again significantly improves the training dynamics, reaching a higher final test accuracy and the best test accuracy of GRPO without LILO in 1.5x fewer steps. Test accuracy is averaged over the 5 datasets: MATH, AIME, AMC, Minerva and Olympiad Bench. The individual training curves for each benchmark are shown in Figure 4, where we see learnability significantly improves performance in all bar AIME, where accuracy is similar to the baseline.

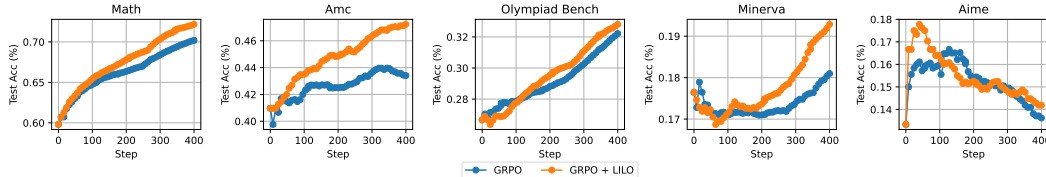

Figure 4: Adding LILO to GRPO increases the model improvement rate on 4/5 unseen test datasets.

## 7 Further analysis

**LILO introduces an interpretable training curriculum** In Figure 5 we plot the average number of reasoning steps in the gold solution to each question trained on at a given iteration. Note that these gold solutions are not used to calculate learnability, but provide a human-interpretable way to see what type of questions are being prioritised during training. For MATH, we initially select simpler questions (fewer steps) but gradually include harder ones. On GSM8K, the model quickly masters the average question, prompting us to focus on more difficult ones.

**Learnability doesn't solve RL's generalisation gap** The results in the previous section show that RL training can increase the test accuracy of LLMs, and learnability helps this happen faster and to a

greater extent. However, the improvements, even with learnability, are modest, with no algorithm allowing Rho-Math-1B or Qwen-2.5-1B to perfect MATH, GSM8K or other datasets. However, looking at training accuracy (full training curves in Appendix F), we see that training accuracy is often nearly 2x higher. In the GSM8K experiments, for example, the training accuracy nears 95% whilst test accuracy stays below 60%. While LILO causes train accuracy to increase faster, the ratio between train and test accuracy is unchanged. We plot this in Figure 10 in Appendix F.

**Finding learnable questions gets harder throughout training** Figure 6 shows that during PPO training on MATH, we remain able to select high learnability batches. For GSM8K, rejection sampling starts to be unable to find high learnability batches. This could be a factor in why PPO closes in on PPO + Learnability later in training on GSM8K, whereas for MATH the two methods continue to diverge. It motivates future work on dynamically choosing $N$ at each iteration based on train accuracy, and generally spending more compute for finding learnable questions later on in training.

**Learnability isn't a silver bullet** We took Oat-Zero-1.5B [22], a state-of-the-art finetune of Qwen-2.5-1.5B [12], and tried to further improve it on its existing dataset by prioritising questions that still had high learnability. Despite being able to find batches of highly learnable questions, performance only minimally improves. This could be due to many reasons - a lack of capacity in the model, entering the overfitting/no generalisation regime, or learning new questions whilst forgetting old ones. We are excited to investigate this further.

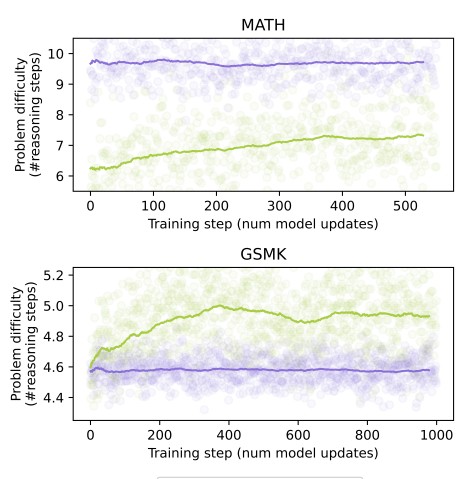

Figure 5: Questions selected by LILO correlate with the number of reasoning steps in the gold standard solution, despite LILO not having access to this information.

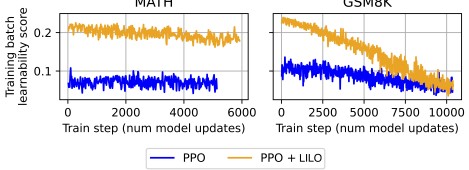

Figure 6: LILO struggles to find high learnability questions towards the end of training. This happens to a lesser degree on MATH than on GSM8K.

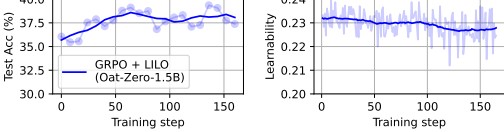

Figure 7: Despite the prevalence of highly learnability training questions, GRPO+LILO only minimally improves Oat-Zero-1.5B. This may be due to the model already having undergone significant post-training.

## 8 Related work

**Curricula with LLMs for RL:** Rho-1b [11] shows the effectiveness of curricula during supervised training of LLMs, but prior to the initial release of this paper, there was minimal work applying curricula to RL on LLMs. While [25] try PLR [26] and backtracking [27], they see no improvement over PPO. DPOP [28] applies various heuristics to improve the diversity of question attempts for use with DPO, but the heuristics are not linked to model performance and do not change during training.

Since the initial release of this paper in February 2025, several works have been released that validate some of our findings. DAPO [29] filters out training on questions that produce all successful or all failing attempts. Kimi-1.5 [30] samples questions proportional to $1 - \hat{p}$, i.e., upweighting difficult questions. "Not all rollouts are useful" [31] trains on a subset of question attempts with the highest variance. They provide no theory to explain why this is useful and present only a single experiment with GRPO on GSM8K. Llama 4 [32] uses a curriculum of increasingly difficult questions and filters out questions with zero advantage. They provide no theory, training code or experiments to validate this choice.

**Unsupervised Environment Design**: A UED problem features a family of MDPs parametrised by a set of values $\theta$, known as a "level" [33, 26]. The goal is to vary the $\theta$ used during training such that performance remains high on unseen or adversarially chosen $\theta$ at test time. The majority of previous work on learnability and UED has been on embodied-agent style robotics tasks, such as Minigrid [34], XLand-Minigrid [35], JaxNav [36] and variants OpenAI's bidpeal walker [37]. However, one can view each question in a reasoning dataset as a "level" of the answer-generation MDP. Viewing RL on LLMs as a UED problem opens up a whole world of literature that could be adapted for LLMs to further improve capabilities. Notably, SFL [18], which inspired early work on this paper, trains on both high learnability and randomly sampled levels throughout training on the robotics tasks above. Having proved Theorem 3.1, we train only on high learnability questions. Many elements of SFL are unsuitable for RL on LLMs: Notably, maintaining a buffer of high learnability questions that is only periodically refreshed leads to massive overfitting in LLMs (see Figure 10 in Appendix F). Sampling is slower for LLMs than in their JAX [38] environments, so we 1) rollout for a fixed number of episodes, rather than timesteps, which allows us to use fast inference engines and 2) reuse samples from learnability estimation in training. For further related work on UED and its connection to LLMs and RL, see Appendix G.

## 9   Limitations and future work

There are many exciting ways to further reduce LILO's sampling overhead. Appendix D.1 details one such way and presents some initial results. As suggested in Section 7, it is likely more efficient to dynamically allocate more compute to finding learnable questions as it gets more difficult later in training. LILO currently discards useful data if more than $|B|$ high learnability questions are identified. Making LILO fully asynchronous, similar to AsyncRLHF [20], could alleviate this. LILO is also stateless - no information from previous steps is used to estimate learnability in the current step. Future work could also explore using the LLM itself to estimate learnability without any actual question attempts. This is similar to some of the work on LLM confidence estimation and calibration [39].

LILO uses learnability to prioritise existing training data. Future work could look at using learnability as a metric to produce new training data at the correct level of difficulty. This could build on the UED algorithm ACCEL [40] for mutating questions, and existing work for automatically producing RLVR tasks [41].

Learnability does not account for inherent aleatoric uncertainty in a question, e.g, "*I just flipped a coin. Is it head or tails?*". Such questions may not be actually *learnable* by the model, despite having high variance of success, and thus high learnability score. There is also no quantification of whether a question is "worth learning", e.g its relevance to the test set.

Learnability in prior work was only considered for binary rewards. Our Theorem 3.1 suggests that more generally, including for non-binary rewards, that maximising learnability, defined as the variance of the final reward, maximises expected model improvement. However, more experiments are needed to empirically verify this. There is a growing body of work looking at unsupervised RL, in which no rewards, or even tasks, are given to the model. In this situation, the agent simply explores to better understand the world around them. It remains to be seen how this concept, previously implemented for vision-transformer or diffusion-based world models, can be translated into the LLM domain.

## 10   Conclusion

In this work, we revisited the formal framing of training LLMs to reason using RL, connecting it to existing theories of learnability, active curriculum learning (ACL), and unsupervised environment design (UED). We contribute new theoretical results showing that prioritising learnability during training maximises expected policy improvement. Building on this, we propose LILO, a simple, practical method for prioritising learnable questions during LLM training, and demonstrate its ease of integration by adding it to two widely used training libraries. We evaluate our approach using three algorithms (GRPO, PPO, and VinePPO) across three reasoning benchmarks (MATH, GSM8K, and ORZ57K). Our results show that LILO not only improves final performance by several percentage points but also requires $3\times$ fewer training steps to exceed the same performance of training without LILO. We conclude by discussing the limitations of current learnability-based methods and outlining future directions toward developing LLM agents capable of autonomously and continually expanding the boundaries of human knowledge.

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

# Supplementary Material

## Table of Contents

# A Policy gradient algorithms

The standard RL framework [42] considers an environment formulated as a Markov Decision Process (MDP) $M = (\mathcal{S}, \mathcal{A}, \mathbb{T}, R, \mu_0, \gamma)$, where $\mathcal{S}$ and $\mathcal{A}$ denote the state and action spaces, $\mathbb{T}(s_{t+1}|s_t, a_t)$ denotes the transition dynamics, $R(s_t, a_t)$ denotes the reward function, $\mu_0$ the initial state distribution and $\gamma \in (0, 1)$ is the discount factor. Given an episode of length $T$, $[s_0, a_0, R_0, s_1, a_1, R_1 \ldots s_{T-1}, a_{T-1}, R_{T-1}]$ with actions sampled by the policy $\pi_\theta(a_t|s_t)$, the goal is to find the policy $\pi_\theta$ that maximizes the expected discounted return:

$$\mathbb{E}_{\mu_0, \pi_\theta, \mathbb{T}} \left[ \sum_{t=0}^{T-1} \gamma^t R_t \right] \tag{2}$$

This objective can be maximized using gradient ascent with the following gradient estimator:

$$\nabla_\theta J(\theta) = \mathbb{E}_{\substack{\mathbf{s} \sim \mathcal{D}, \\ \mathbf{a} \sim \pi_\theta(\cdot|\mathbf{s}_t)}} \left[ \sum_{t=0}^{T-1} \nabla_\theta \log \pi_\theta \left( a_t \mid s_t \right) R_t \right] \tag{3}$$

For a lower variance gradient estimator, the reward $R_t$ can be replaced with the 'advantage' $A_t$, which quantifies how much better taking action $a_t$ in state $s_t$ is compared to the average action according to the current policy $\pi_\theta$. Given a learning rate $\beta$, the policy parameters are updated iteratively with:

$$\theta_{k+1} = \theta_k + \beta \nabla_\theta J(\theta_k) \tag{4}$$

We now consider applying this RL framework to question-answering in LLMs, denoting the sequence of tokens that make up the question and LLM response as $\mathbf{x} = x_{0:n}$ and $\mathbf{y} = y_{0:T}$ respectively. There are multiple different treatments of the RL framework for LLMs. The simplest view treats response generation as 'contextual bandits' and to consider the question to be the initial state $s_0 = x_{0:n}$, the generation of an entire response to be a single action $a_0 = y_{0:T}$, and thus the horizon to be $T = 1$. An alternative is to view response generation as a token-level MDP, so the state is the sequence of previous question and response tokens $s_t = [\mathbf{x}, y_{0:t-1}]$, the action is the next token in the response $a_t = y_t$, and the transition function is deterministic and simply involves concatenating the new token $y_t$ onto the previous state to get $s_{t+1} = [\mathbf{x}, y_{0:t}]$. In RL applied to LLMs a single reward $R$ is typically only received after the whole response has been generated. Therefore with on-policy data the $T = 1$ contextual bandits treatment and the token-level treatment with $\gamma = 1$ are equivalent.

In deep RL the advantage is typically estimated as $A(a_t, s_t) = R(a_t, s_t) - V_\phi^{\pi_\theta}(s_t)$, where $V_\phi^{\pi_\theta}(s_t)$, an additional learned value baseline, is subtracted from the reward. In LLMs, however, fitting value functions has proven tricky and instead a sample-based approach involving sampling multiple responses for each prompt has become popular. There are several different popular expressions for advantage. RLOO [15] samples $G$ responses $\{\mathbf{y}^{(i)}\}_{i=1}^G$ per question and uses $A^{(i)} = R^{(i)} - \frac{1}{G-1} \sum_{j \neq i}^G R^{(j)}$. Similarly, GRPO makes use of $G$ sampled responses per prompt, instead using $A^{(i)} = (R^{(i)} - \text{mean}_j R^{(j)}) / \text{std}_j R^{(j)}$[1].

This is generally applied token-wise, resulting in the following gradient estimator:

$$\nabla_\theta J(\theta) = \mathbb{E}_{\substack{\mathbf{x} \sim \mathcal{D}, \\ \mathbf{y}^{(i)} \sim \pi_\theta(\cdot|\mathbf{x})}} \left[ \sum_{i=1}^G \sum_{t=0}^T \nabla_\theta \log \pi_\theta \left( y_t^{(i)} \mid \mathbf{x}, y_{0:t-1}^{(i)} \right) A^{(i)} \right] \tag{5}$$

While RLOO and GRPO assign the same advantage $A$ for all response tokens, VinePPO [5] aims for fine-grained credit assignment. Like RLOO and GRPO, they again use a sample-based approach instead of relying on a learned value network, and estimate $V^{\pi_\theta}(s_t = [\mathbf{x}, y_{0:t}])$ by sampling multiple completions $y_{t+1:T}$ onwards using the current policy $\pi_\theta$ and averaging their final rewards. The advantage is then calculated as $A_t = V^{\pi_\theta}(s_{t+1}) - V^{\pi_\theta}(s_t)$, where a sample-based estimate is used for $V^{\pi_\theta}$. This approach avoids the difficulties of training a separate value network for LLMs while still enabling token-level credit assignment.

PPO style clipping is often applied to increase sample efficiency by enabling use of off-policy data.

---

[1]The GRPO advantage has been found to perform better without the division by $\text{std}_j R^{(j)}$[4].

# B Proof of Theorem 3.1

*Proof.* We see that

$$\|\nabla_\theta J(\theta)\|^2 = \left\| \mathbb{E}_{\pi_\theta} \left[ \sum_{t=0}^{T-1} \nabla_\theta \log \pi_\theta(a_t|s_t) A_t \right] \right\|^2$$

$$\leq \mathbb{E}_{\pi_\theta} \left[ \left\| \sum_{t=0}^{T-1} \nabla_\theta \log \pi_\theta(a_t|s_t) A_t \right\|^2 \right]$$

$$= \sum_{t,t'=0}^{T-1} \mathbb{E}_{\pi_\theta} \left[ \nabla_\theta \log \pi_\theta(a_t|s_t)^T \nabla_\theta \log \pi_\theta(a_{t'}|s_{t'}) A_t A_{t'} \right]$$

$$= \sum_{t,t'=0}^{T-1} \mathbb{E}_{\pi_\theta} \left[ \nabla_\theta \log \pi_\theta(a_t|s_t)^T \nabla_\theta \log \pi_\theta(a_{t'}|s_{t'}) \right] \mathbb{E}_{\pi_\theta} \left[ A_t A_{t'} \right],$$

where in the second inequality we have used Jensen's inequality, and in the last equality we have used the assumption that $\left( \nabla_\theta \log \pi_\theta(a_t|s_t) \right)^T \nabla_\theta \log \pi_\theta(a_{t'}|s_{t'})$ and $A_t A_{t'}$ are uncorrelated for all $t, t' = 0, ..., T-1$. This proves the statements for REINFORCE, where $A_t = r(s_T)$, and RLOO, where $A_t = r(s_T) - \mathbb{E}_{\pi_\theta} [r(s_T)]$. For PPO and VinePPO, where $A_t = V^{\pi_\theta}(s_{t+1}) - V^{\pi_\theta}(s_t)$, we now show that $\mathbb{E}_{\pi_\theta} [A_t A_{t'}] = 0$ for $t \neq t'$, and $\mathbb{E}_{\pi_\theta} [A_t^2] = \mathbb{E}_{\pi_\theta} [V^{\pi_\theta}(s_{t+1})^2 - V^{\pi_\theta}(s_t)^2]$. Together with the fact that $r(s_T) = V^{\pi_\theta}(s_0)$ and $\mathbb{E}_{\pi_\theta} [r(s_T)] = V^{\pi_\theta}(s_0)$, this will finish the proof of the theorem.

We see that

$$\mathbb{E}_{\pi_\theta} \left[ (V^{\pi_\theta}(s_{t+1}) - V^{\pi_\theta}(s_t))^2 \right]$$

$$= \mathbb{E}_{\pi_\theta} \left[ \left( V^{\pi_\theta}(s_{t+1})^2 - 2V^{\pi_\theta}(s_{t+1})V^{\pi_\theta}(s_t) + V^{\pi_\theta}(s_t)^2 \right) \right]$$

$$= \mathbb{E}_{\pi_\theta} \left[ \left( V^{\pi_\theta}(s_{t+1})^2 - 2\mathbb{E}_{\pi_\theta} [V^{\pi_\theta}(s_{t+1})V^{\pi_\theta}(s_t))|s_t] + V^{\pi_\theta}(s_t)^2 \right) \right]$$

$$= \mathbb{E}_{\pi_\theta} \left[ \left( V^{\pi_\theta}(s_{t+1})^2 - 2V^{\pi_\theta}(s_t) \sum_{a_t} \pi_\theta(a_t|s_t) \sum_{s_{t+1}} \mathcal{T}(s_{t+1}|s_t, a_t)V^{\pi_\theta}(s_{t+1}) + V^{\pi_\theta}(s_t)^2 \right) \right]$$

$$= \mathbb{E}_{\pi_\theta} \left[ \left( V^{\pi_\theta}(s_{t+1})^2 - 2V^{\pi_\theta}(s_t)^2 + V^{\pi_\theta}(s_t)^2 \right) \right]$$

$$= \mathbb{E}_{\pi_\theta} \left[ \left( V^{\pi_\theta}(s_{t+1})^2 - V^{\pi_\theta}(s_t)^2 \right) \right],$$

$$(6)$$

where $\mathcal{T}(s_{t+1}|s_t, a_t)$ is the transition function of the underlying MDP. We now turn to the off-diagonal terms, where without loss of generality we let $t > t'$:

$$\mathbb{E}_{\pi_\theta} \left[ (V^{\pi_\theta}(s_{t+1}) - V^{\pi_\theta}(s_t))(V^{\pi_\theta}(s_{t'+1}) - V^{\pi_\theta}(s_{t'}))) \right]$$

$$= \mathbb{E}_{\pi_\theta} \left[ \mathbb{E}_{\pi_\theta} \left[ (V^{\pi_\theta}(s_{t+1}) - V^{\pi_\theta}(s_t))(V^{\pi_\theta}(s_{t'+1}) - V^{\pi_\theta}(s_{t'}))|s_0, ..., s_t] \right] \right]$$

$$= \mathbb{E}_{\pi_\theta} \left[ (\mathbb{E}_{\pi_\theta} [(V^{\pi_\theta}(s_{t+1})|s_t] - V^{\pi_\theta}(s_t)) (V^{\pi_\theta}(s_{t'+1}) - V^{\pi_\theta}(s_{t'}))) \right]$$

$$= \mathbb{E}_{\pi_\theta} \left[ \left( \sum_{a_t} \pi_\theta(a_t|s_t) \sum_{s_{t+1}} \mathcal{T}(s_{t+1}|s_t, a_t)V^{\pi_\theta}(s_{t+1}) - V^{\pi_\theta}(s_t) \right) (V^{\pi_\theta}(s_{t'+1}) - V^{\pi_\theta}(s_{t'})) \right]$$

$$= \mathbb{E}_{\pi_\theta} \left[ (V^{\pi_\theta}(s_t) - V^{\pi_\theta}(s_t)) (V^{\pi_\theta}(s_{t'+1}) - V^{\pi_\theta}(s_{t'})) \right]$$

$$= 0.$$

$$(7)$$

$\square$

## C  Significance tests

| | GSM8K | | MATH | |
|---|---|---|---|---|
| | **Corr** | **Sig** | **Corr** | **Sig** |
| REINFORCE: No correlation between $r(s_T)^2$ and $\|\sum_{t=0}^{T-1} g_t\|^2$ | -0.0460 | Yes | -0.0315 | Yes |
| RLOO: No correlation between $(r(s_T) - \mathbb{E}[r(s_T)])^2$ and $\|\sum_{t=0}^{T-1} g_t\|^2$ | -0.0169 | Yes | -0.0211 | Yes |
| GRPO: No correlation between Learnability and $\|\sum_{t=0}^{T-1} g_t\|^2$ | -0.0327 | Yes | -0.0319 | Yes |

Table 3: Since the underlying MDP in Theorem 3.1 depends on the question on which we train, it is in principle possible that the quantities $\mathbb{E}_{\pi_\theta}\left[\|\nabla_\theta \log \pi_\theta(a_t|s_t)\|^2\right]$, $t = 0, ..., T-1$, depend on the learnability of that question. In particular, they could decrease with learnability, such that the upper bound for the gradient magnitude as a whole could decrease with learnability. However, while the formal MDP changes, the weights of the underlying LLM do not depend on the question that is chosen to train. Thus we assume that in practice, those quantities do not depend significantly on the learnability of the current question. This table tests this assumption empirically, and shows no statistically significant correlation between learnability and $\mathbb{E}_{\pi_\theta}\left[\|\nabla_\theta \log \pi_\theta(a_t|s_t)\|^2\right]$, for any $t = 0, ..., T-1$.

| **Assumption** | GSM8K | | MATH | |
|---|---|---|---|---|
| | **Corr** | **Sig** | **Corr** | **Sig** |
| No correlation between $g_t \cdot g_{t'}$ and $[V(s_{t+1}) - V(s_t)][V(s_{t'+1}) - V(s'_t)]$ | 0.0357 | No | 0.0132 | No |
| No correlation between $\|g_t\|^2$ and $V(s_{t+1}) - V(s_t)$ | -0.0025 | No | -0.0030 | No |
| No correlation between $\|g_t\|^2$ and $t$ | -0.0072 | No | -0.0046 | No |
| No correlation between $\|g_t\|^2$ and learnability | -0.0057 | No | 0.0105 | No |

Table 4: This table checks the assumptions required by the proof of Theorem 3.1

| | GSM8K | | MATH | |
|---|---|---|---|---|
| **Learning algorithm** | **Corr** | **Sig** | **Corr** | **Sig** |
| REINFORCE | 0.1629 | Yes | 0.3145 | Yes |
| RLOO | 0.1485 | Yes | 0.4262 | Yes |
| GRPO (normalised) | 0.1293 | No | 0.3842 | Yes |
| GRPO (unnormalised) | 0.1690 | Yes | 0.4178 | Yes |
| PPO | 0.3786 | Yes | 0.7140 | Yes |
| VinePPO | 0.3786 | Yes | 0.7140 | Yes |

Table 5: This table tests Theorem 3.1 end to end, by showing the correlation between learnability and $\|\nabla_\theta J(\theta)\|^2$

# D Additional method

## D.1 An algorithm for selecting high learnability questions with no additional sampling cost

Algorithm 2 uses rejection sampling to get batches of high learnability questions. It uses $|D| \times N_{\text{learnability}}$ rollouts to find $|B|$ high learnability questions and generate a batch of $|B| \times N_{\text{learnability}}$ question attempts to learn from. This is in contrast to uniform sampling, which uses only $|B| \times N_{\textbf{learnability}}$ samples.

As discussed in Section 4, in many situations this is practical and results in a negligible increase in wall clock time. However, for situations where this is not the case, ie sampling is extremely slow and cannot be parallelised, we present Algorithm 3 which is smarter in how it samples.

Algorithm 3 starts in the same way as Algorithm 2, sampling a candidate pool of $|D|$ questions from the dataset. However, unlike Algorithm 2, it only samples 2 attempts for each question in this pool. Algorithm 3 then proceeds to iteratively estimate the learnability of each question in the pool, sample a promising high learnability question, and draw 2 further samples from it. At each iteration, the new samples further refine the learnability estimates. This is a smarter way to allocate a given sampling budget, $N$.

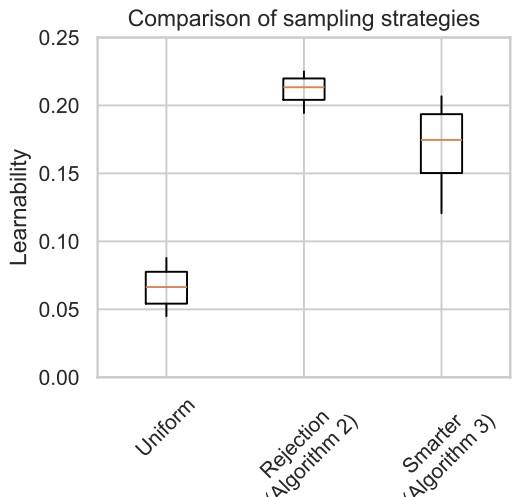

Figure 8: The smarter sampling used by Algorithm 3 produces high learnability batches comparable to rejection sampling, but using $4\times$ fewer samples. It uses the same number of samples as uniform sampling, ie LILO + Smarter has no additional sampling overhead.

We show the learnability of batches generated by Algorithm 3 in Figure 8. We see that uniform sampling consistently produces low learnability batches, and, as could be expected, the rejection sampling approach consistently produces high learnability batches. However, despite using $4\times$ fewer samples than rejection sampling, the smarter sampling used by Algorithm 3 produces batches with nearly as high learnability. Algorithm 3 produces much high learnability batches than uniform sampling despite using the same sampling budget.

---

**Algorithm 3** Get Learnable Questions Using Smarter Sampling

---

    **Input:** Model parameters, $\theta$
           Size of question batch to return, $|B|$
    **Hyperparameters:** Size of candidate pool, $|D|$
                   Total number of question rollouts, $N$
1: Sample candidate pool of $|D|$ questions from dataset
2: Rollout 2 attempts per question
3: $i = 2 \times |D|$
4: **while** $i < N$ **do**
5:     Compute success rates for each question $= \hat{p}$
6:     Compute learnability for each question $= \hat{p}(1 - \hat{p})$
7:     Select question from candidate pool with probability proportional to its learnabiliity
8:     Rollout 2 more attempts for that question
9:     $i = i + 2$
10: **end while**
11: Return all $N$ generated attempts

---

# E  Additional experimental setup details

## E.1  Alterations to VinePPO training setup

We increased the number of gradient accumulation steps and added Deepspeed ZeRO stage 2 to allow for training on a single 48GB Nvidia L40s GPU and thus run multiple experiments in parallel on an 8 * 48GB node. This did not change the overall effective batch size of 512 episodes (64 levels * 8 rollouts per level) and thus our training dynamics are identical to VinePPO.

## E.2  Compute resources

The PPO and VinePPO experiments took  1 week on 4xL40s GPUs for each algorithm, dataset combination. The GRPO experiments took  1 day on 8xH200 GPUs for each of GRPO with LILO and GRPO without LILO.

# F  Additional results

Table 6: Test accuracy@1 for 4 different training runs on the MATH dataset. CollegeMath and OlympidBench were not seen during training. **LILO consistently improves generalisation.**

|  | COLLEGE MATH (%) | OLYMPIAD BENCH (%) |
|---|---|---|
| ORIGINAL SFT | 20.3 | 2.6 |
| PPO | 25.4 | 3.5 |
| **PPO + LILO** | **26.4** | **3.8** |
| VINEPPO | 26.9 | 4.1 |
| **VINEPPO + LILO** | **28.8** | **4.5** |

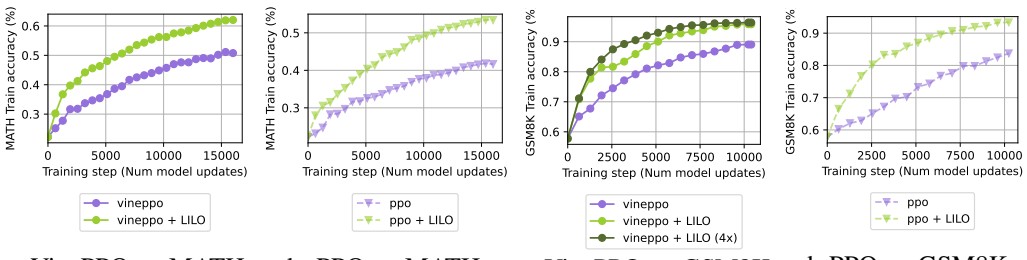

a: VinePPO on MATH     b: PPO on MATH     c: VinePPO on GSM8K     d: PPO on GSM8K

Figure 9: Train accuracy when training with and without LILO. For a full discussion of experimental setup see section 5. **In all scenarios train accuracy increases faster with LILO.**

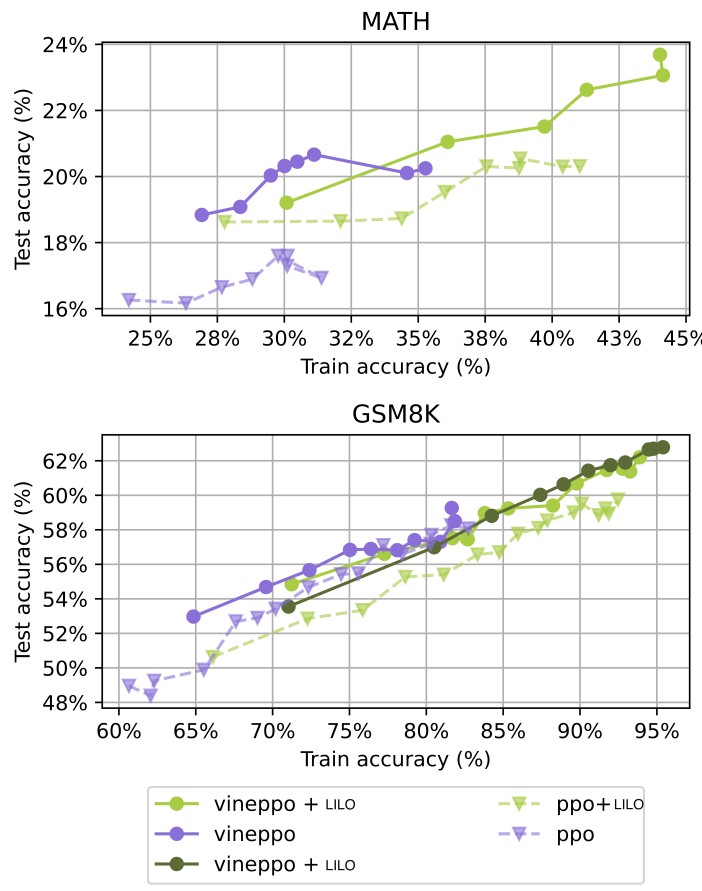

Figure 10: Train accuracy vs test accuracy shows overfitting and generalisation of different runs. **Despite LILO's higher train accuracy, all runs fall roughly on the same line, indicating the same level of generalisation and degree of overfitting**.

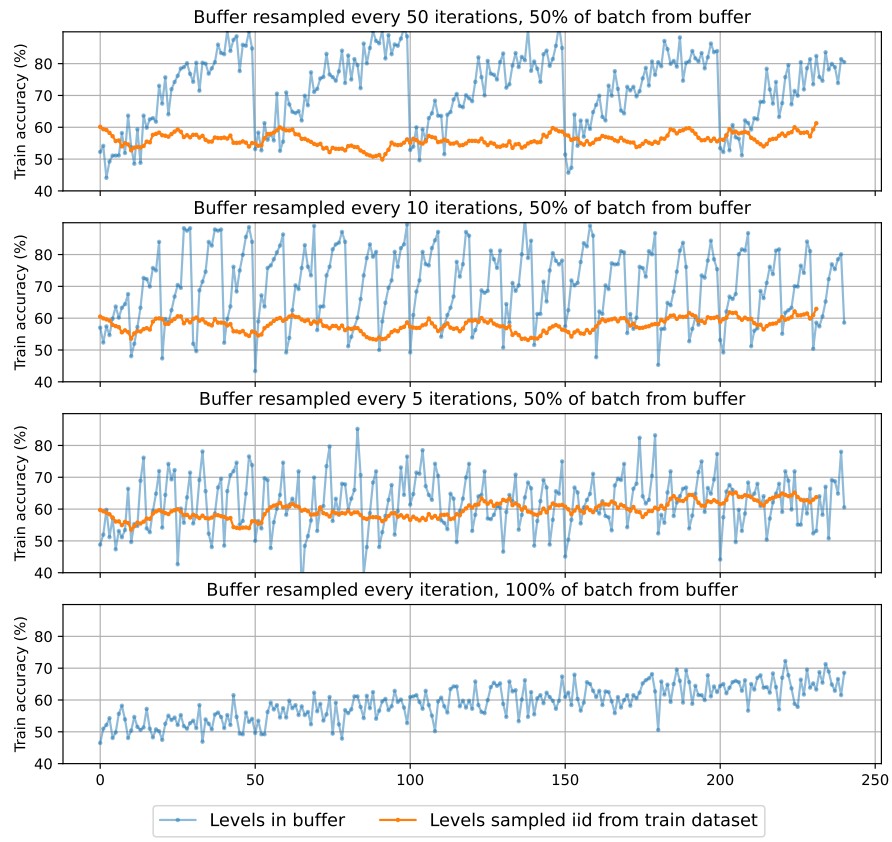

Figure 11: Exploring finding learnable questions less frequently, as suggested by [18]. 50% of the batch is randomly sampled each iteration, and the rest of the batch is sampled from a buffer of high learnability questions found by LILO. The model overfits to the high learnability questions when the buffer is refreshed every $50, 10, 5$ steps. Only when updated every step did we find overfitting stopped and generalisation started to occur.

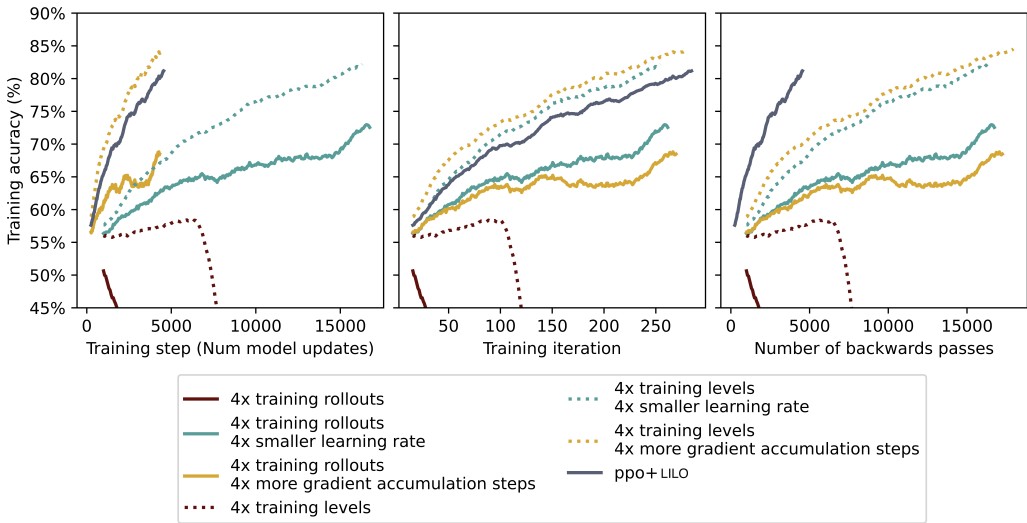

Figure 12: Training on all generated samples vs training on the top-K highest learnability samples. We compare generating the additional samples from 4x more levels, or 4x more rollouts per level. We use the additional samples 1) naively, by increasing the number of batches, 2) increasing the number of batches but decreasing the learning rate 4x and 3) increasing the effective batch size by doing 4x the gradient accumulation steps. Whilst training on 4x the samples can challenge the performance of PPO+LILO, it is significantly slower due to the outsized cost of doing more model forward/backwards passes during training.

# G  Further related work

We now cover various framings of the language generation MDP discussed in Section 2 and their related work on Automated Curriculum Learning (ACL).

## G.1  ACL in Supervised Learning and Bandit Problems

Choosing a question to train on during each step can be seen as a bandit problem. The seminal work [43] provides an overview of ACL in this setting. The central idea is to define a utility measure $U(q)$ for each question or training sample and then design a curriculum that sequences the data accordingly. Early methods [44] used human-designed heuristics for $U$, progressing training from the easiest to the hardest examples. More advanced ACL methods in this setting can be loosely categorized by how they define $U$, such as:

1. **Loss-Based Methods**: Many methods use loss to prioritize training on hard data points [45, 46, 47]. In [48] the authors use the loss from a pre-trained model to estimate the difficulty of new samples for a freshly initialized network learning a new task.

2. **Uncertainty or Entropy**: Several papers use the entropy of the answer distribution to prioritize training on data points the model is "unsure" about [49, 50, 51]. **LILO can be seen as using return variance—or learnability—as an estimator of entropy or uncertainty**. For Bernoulli random variables, as in the reasoning setting, maximum entropy corresponds to maximum variance.

3. **Information Gain or Learning Progress**: In some settings, it is possible to estimate or empirically compute the effect of training on a data point. This allows prioritizing samples that maximize the change in loss—i.e., the model's learning progress [52, 11, 53, 49]. One can also aim to maximize the change in entropy or information gain [49, 53]. These approaches are challenging to apply at the scale of modern LLMs.

4. **Gradient-Based Approaches**: One can select data points that minimize the variance of the gradient estimator computed by SGD [54, 55].

Self-paced learning [56] is an early approach that allows the model to determine the pace at which it incorporates harder examples with higher values of $U$. [57, 58] introduced the concept of a teacher–student setup, where the teacher is trained to optimize the student's learning.

## G.2  ACL in Multi-Task Reinforcement Learning

Each question in a training dataset can be thought of as defining its own MDP, with the same state and action spaces but a unique reward function encoding the question's answer. This framing is equivalent to multi-task RL (MTRL) [59], where each question constitutes a separate task.

ACL work for MTRL roughly follows the categories above. [60] uses the entropy of the value function to prioritize training on tasks with the highest uncertainty. [61] selects training tasks based on their information gain relative to test tasks. Similarly, [62] prioritizes tasks that maximize entropy gain.

## G.3  ACL in Goal-Conditioned RL

Since the MDP defined by each question shares the same state and action spaces, a question can be viewed as a "goal" for the agent. [63] samples goals using a learned goal distribution that maximizes entropy. [64] learns which goals are teachable based on an agent's competence. It is intriguing to consider whether ideas from [65]—in which a GAN is trained to generate goals of appropriate difficulty—could be applied to reasoning with LLMs.

## G.4  Unsupervised Environment Design

Formally, the language generation MDP is extended to use the parameter $\theta$. Each $\theta \in \Theta$ corresponds to a question from a training dataset, and the UED adversary will determine the distribution over which $\theta$ is sampled from $\Theta$. Given a choice of $\theta$, the starting distribution for each level $p_\theta$ is just

the question $\mathbf{x}$, and the reward function $R_\theta$ computes accuracy of the generation $\mathbf{y}$. The transition function is unchanged.

There are some key practical differences between the LLM reasoning setting and the setting UED was originally designed for. SFL was previously employed on simple, fast, vectorised environments such as Minigrid, XLand-Minigrid and JaxNav. These environments have millions of unique levels. GSM8K, on the contrary, is just 8,000 questions. Models in the traditional UED setting are small, perhaps 1M parameters. Whilst in this work we looked at a 1B parameter model, Rho-Math-1B [11], LLMs with far more parameters have been trained with RL, such as the 671B parameter Deepseek-R1 [1]. SFL has therefore previously been employed in settings where each training batch contains experience from 1000s of levels and 100s of episodes per level, even on a single GPU. Mention how many rollouts per question is done in Deepseek, Tulu, RLHF etc In contrast in VinePPO, they train on a batch of just 8 rollouts for each of 64 levels.

Whilst the increase in popularity of JAX means that some popular RL environments are now vectorised, the sampling of trajectories remains expensive and a major bottleneck for many RL tasks. In many LLM training setups however, sampling new trajectories is far more scalable than model updates, as trajectory collection can be distributed across independent processes that require no inter-process communication. This sampling computation can be fully pipelined and leverages highly optimized specialized inference engines like vLLM. Furthermore, since trajectory sampling only involves forward passes, it eliminates the need to store activations or optimizer states, making it significantly more memory-efficient than training updates.

Prioritised level replay (PLR) [26] generates random levels, samples trajectories, and adds high scoring levels to a buffer. TD-error is typically used as the score function. ACCEL [40] extends this with a mechanism to mutate previously high-scoring levels, to generate new levels that train the agent on the frontier of its capabilities. PAIRED [33] co-trains a level-selecting adversary and two agents, a protagonist and an antagonist. It aims to maximise regret by maximising the difference in performance between the protoganist and antogonist. SFL [18] discards the notion of regret, instead using learnability to select which levels to replay. SFL inspired early work on this paper, but significant changes we required to transfer it to LLMs. Namely we had to compute learnability at every step, using a fixed attempt budget instead of timestep budget (more suitable for inference engines like vLLM) and reuse trajectories generated during learnability estimation. This results in an overall simpler, cleaner algorithm to that used in SFL. Our method is also similar to Prioritised Experience Replay (PER) [66], which uses TD-error prioritisation instead of learnability prioritisation.

# H    Examples of high and low learnability questions

**Dataset: GSM8K**

- **Question:** Alton owns a business. He is currently renting a space that costs $20 per week. If Alton earns $8 per day, how much is his total profit every week?

  - **Gold Solution:**
    * Calculate weekly earnings: $8 $\times$ 7 = $56
    * Subtract weekly rent: $56 - $20 = $36
    * **Answer:** 36
  - **Number of Reasoning Steps:** 3
  - **Success Rate ($p$) at $t = 0$:** 1.0
  - **Learnability ($p(1-p)$) at $t = 0$:** 9

- **Question:** Mrs. Smith wanted to buy items worth $500. She went to a boutique with the $500 but realized she would need two-fifths more money than she had. If the shop owner gave her a discount of 15%, how much more money will she still need?

  - **Gold Solution:**
    * Calculate additional money needed: $\frac{2}{5} \times 500 = 200$
    * Total cost before discount: $500 + $200 = $700
    * Calculate discount: $0.15 \times 700 = 105$
    * Subtract discount from total cost: $700 - $105 = $595
    * Additional money needed: $595 - $500 = $95
    * **Answer:** 95
  - **Number of Reasoning Steps:** 6
  - **Success Rate ($p$) at $t = 0$:** 0.5
  - **Learnability ($p(1-p)$) at $t = 0$:** 0.25

- **Question:** In a fruit salad, there are raspberries, green grapes, and red grapes. There are seven more than 3 times the number of red grapes as green grapes. There are 5 fewer raspberries than green grapes. If there are 102 pieces of fruit in the salad, how many red grapes are in the salad?

  - **Gold Solution:**
    * Let $G$ represent the number of green grapes.
    * Red grapes: $3G + 7$
    * Raspberries: $G - 5$
    * Total fruit equation: $G + (3G + 7) + (G - 5) = 102$
    * Simplify: $5G + 2 = 102$
    * Solve for $G$: $5G = 100 \Rightarrow G = 20$
    * Calculate red grapes: $3 \times 20 + 7 = 67$
    * **Answer:** 67
  - **Number of Reasoning Steps:** 10
  - **Success Rate ($p$) at $t = 0$:** 0
  - **Learnability ($p(1-p)$) at $t = 0$:** 0

**Dataset: MATH**

- **Question:** Suppose $p(x)$ is a monic cubic polynomial with real coefficients such that $p(3 - 2i) = 0$ and $p(0) = -52$. Determine $p(x)$ (in expanded form).

  - **Gold Solution:**
    * Since $p(x)$ has real coefficients and $3 - 2i$ is a root, its complex conjugate $3 + 2i$ is also a root.

* The quadratic factor with roots $3 - 2i$ and $3 + 2i$ is:

$$(x - (3 - 2i))(x - (3 + 2i)) = (x - 3 + 2i)(x - 3 - 2i)$$
$$= (x - 3)^2 - (2i)^2$$
$$= x^2 - 6x + 9 + 4$$
$$= x^2 - 6x + 13$$

* Since $p(x)$ is cubic, it can be expressed as $p(x) = (x^2 - 6x + 13)(x - r)$.
* Given $p(0) = -52$:

$$p(0) = (0^2 - 6 \cdot 0 + 13)(0 - r)$$
$$-52 = 13(-r)$$
$$r = 4$$

* Therefore, $p(x)$ is:

$$p(x) = (x^2 - 6x + 13)(x - 4)$$
$$= x^3 - 4x^2 - 6x^2 + 24x + 13x - 52$$
$$= x^3 - 10x^2 + 37x - 52$$

* **Answer:** $x^3 - 10x^2 + 37x - 52$
  – **Number of Reasoning Steps:** 29
  – **Success Rate ($p$) at $t = 0$:** 0
  – **Learnability ($p(1 - p)$) at $t = 0$:** 0

• **Question:** If the sum of the squares of nonnegative real numbers $a$, $b$, and $c$ is 13, and $ab + bc + ca = 6$, then what is the sum of $a$, $b$, and $c$?
  – **Gold Solution:**
    * Start with the identity: $(a + b + c)^2 = a^2 + b^2 + c^2 + 2(ab + bc + ca)$
    * Substitute the given values:

$$(a + b + c)^2 = 13 + 2 \times 6 = 13 + 12 = 25$$

    * Take the square root of both sides:

$$a + b + c = \pm\sqrt{25} = \pm 5$$

    * Since $a$, $b$, and $c$ are nonnegative, $a + b + c$ must be nonnegative.
    * **Answer:** 5
  – **Number of Reasoning Steps:** 5
  – **Success Rate ($p$) at $t = 0$:** 0.5
  – **Learnability ($p(1 - p)$) at $t = 0$:** 0.25

