# OpenReview forum: "LILO: Learning to Reason at the Frontier of Learnability"
_NeurIPS.cc/2025/Conference — NeurIPS 2025 poster_

### Official Review · Reviewer_C2Lj · 2025-06-30

**Clarity:** 3
**Significance:** 4
**Originality:** 4
**Rating:** 5
**Confidence:** 3

**Summary:**

The work studies the learnability of examples in RL and found that learning examples that are not too hard or too easy are the most important.  The work theoretically shows that the expected policy improvement increases with the variance of the final reward.  Motivated by this finding, the proposed LILO framework selects the examples based on learnability and can reach the comparable performance with fewer training steps when applied to varied RL training methods.

**Questions:**

How is N learnability decided?
What is the parameter setting (e.g., temperature, top p) for decoding during sampling?

**Ethical Concerns:**

["NO or VERY MINOR ethics concerns only"]

**Final Justification:**

Thanks for the authors' response. I maintain my score.

**Limitations:**

yes

**Quality:**

3

**Strengths And Weaknesses:**

Strength:
The paper is very well-structured, and the writing is smooth.

The finding reveals examples that are not too hard or too easy are more important to RL, which is very interesting.  The finding is theoretically obtained and evidenced by strong empirical results. The insight is clear and compelling, with promising implications for improving data curation in training LLMs more efficiently.

The proposed method is well-motivated and simple but effective.

Weakness:
One weakness is N_learnability in practice may need tuning to find the most suitable one. Different models may have different N_learnability. It could also depend on the setting during sampling, e.g., top_p, temperature.  Ablation study on N_learnability will be beneficial to show that LILO is robust to different choices of N_learnability.

But the work is generally solid. The core theoretical finding is well-supported by experiments.

---

> ### Author Rebuttal · Authors · 2025-07-31
>
> We thank the reviewer for their time and valuable feedback that has helped us improve this work. We are encouraged by the positive comments and score. We address the suggestion of an additional ablation below and hope this resolves any of the outstanding concerns.
>
> **Tuning N_learnability:**
>
> This is a great point, and different models and hyperparameters may have different optimal N_learnability. Sadly, we don’t have the compute resources to complete a full training run for a variety of choices of N_learnabilty. However, a simple approach to finding N_learnability at the start of training could be to sample a batch of questions, and plot how the choice of N_learnability affects the overall learnability of the selected batches. We do this for GSM8K and MATH below:
>
> |       | N_Learnability | Learnability |
> |-------|----------------|--------------|
> | GSM8K | 1              | 0.000        |
> |    | 2              | 0.145        |
> |       | 4              | 0.195        |
> |       | 8              | 0.221        |
> |       | 16             | 0.231        |
> |       | 32             | 0.238        |
> |       | 64             | 0.241        |
> |       | 100            | 0.243        |
> |       |                |              |
> | MATH  |   1            | 0.000        |
> |       |   2            | 0.107        |
> |       |   4            | 0.134        |
> |       |   8            | 0.169        |
> |       |  16            | 0.175        |
> |       |  32            | 0.182        |
> |       |  64            | 0.183        |
> |       | 100            | 0.186        |
>
> N_learnability=4 or 8 seem to be a nice balance between high learnability batches and low sampling costs.
>
>
> We hope this adequately addresses the issue highlighted and invite any more feedback or points you believe are significant.

---

### Official Review · Reviewer_Qkcn · 2025-06-30

**Clarity:** 3
**Significance:** 2
**Originality:** 2
**Rating:** 4
**Confidence:** 3

**Summary:**

The paper proposes a sample selection mechanism that chooses the examples that are most learnable at the moment (i.e. produces the most variance in the success outcome) as the training batch. The paper finds theoretical relations between learnability and the expected gain of performance. Empirically, the paper finds that the sample selection method improves LLM reasoning on MATH, GSM8K, and ORZ57K datasets by about 5-10% compared to baselines.

**Questions:**

- I requested the wall clock time training curves. But I can't promise if there is a score increase given my other concerns.

**Ethical Concerns:**

["NO or VERY MINOR ethics concerns only"]

**Final Justification:**

The work builds on an interesting relation between "learnability" and LLM reasoning, and applies a sample selection mechanism. While the angle is interesting, the gain in the current experiments is modest. Therefore I recommend a weak accept.

**Limitations:**

The limitation section (Section 9) seems sufficient.

**Quality:**

3

**Strengths And Weaknesses:**

Strengths:
- The paper shows consistent positive improvement (5-10%) over baselines by introducing their curriculum learning component, over 3 different datasets.
- The authors made an effort showing theoretical insights on how learnability equates to policy improvement.

Weaknesses:
- Limited novelty: “Learnability” as a curriculum learning criterion has been proposed in [16], and this paper is just an application of learnability to LLM reasoning. Does learnability as a sample selection criterion benefit particularly to LLM reasoning? This is unclear. Therefore, the novelty is limited.
- Definition of learnability: Although learnability has been proposed before, I have objection to the naming. Higher “learnability” simply means that the model responses are variable, but it doesn’t necessarily mean that they are more learnable. It is possible that although the model uniformly makes the same mistake on the current iteration, the sample is more learnable and after a few updates the model will fix the mistake.
- Theoretical depth: Section 3 discusses how optimizing learnability can lead to policy improvement. However, the insight is limited, and the combination of a theoretical perspective on general RL and an application focus on LLM reasoning is strange.
- Wall clock time comparison: Since the proposed method needs to sample a batch of rollouts for many examples and only choose to update the ones with the top learnability, it can be the case that it takes longer to train. While the paper has compared the timing, it would be good to show the training curve with wall clock time as the horizontal axis, and see if baselines can benefit from prolonged training in the main table.
- Sampling from model policy: Taking the variance over model policy can lead to learning stagnation if the model has no good insight on how to solve the problem with their samples. Furthermore, it can also lead to ignoring certain trajectories in the training environment.

---

> ### Author Rebuttal · Authors · 2025-07-31
>
> We thank the reviewer for their time and valuable feedback that has helped us improve this work. We are encouraged by the positive comments around the improvement that prioritising learnability brings to training LLMs to reason with reinforcement learning. We address the weaknesses and questions one-by-one below and hope this resolves any of the outstanding concerns and that the reviewer will consider revising their rating.
>
> **Originality:**
> - There is a rich history of curricula for traditional RL settings, and as you note, Learnability has been proposed before, such as in [16].
> - We emphasise that the relevance of [16] on reasoning with LLMs is extremely small - there is no clear method for computing and prioritising learnability, there is no suggestion that this should work beyond the REINFORCE algorithm, and is it considered in traditional RL tasks.
> -  Beyond [16], it is non-trivial to decide which curricula methods from traditional RL are suited for the LLM reasoning setting:for example Havrilla tried some and these failed. This is true for other areas of RL too, for example value networks, which are popular in deep RL, have been shown to not transfer to the LLM setting.
> - We identified that Learnability based curricula are particularly well suited for the LLM setting because it is non-stochastic, and uses sparse, binary rewards. Even once we identified this, there was still work to do to adapt it to the LLM setting. For example, in Appendix  E Figure 11 we detail how the frequency with which you recalculate learnable questions, which is typically done every 10-50 timesteps in the traditional setting, is catastrophic in the LLM setting.
> - Crucially, this results in LILO being the first to show that a curriculum method can boost overall performance on LLM reasoning tasks. Given the popularity and resource consumption of this setting this is highly impactful and useful to the community.
>
> **Naming of learnability:**
> - We completely agree that “learnability” is a loaded phrase, and ideally a different term, such as "reward variance" would be used. However there is a large body of previous work (including [16,17,18]) that terms this ‘learnability’ and as such, we must use this term. There are proper mechanisms to attempt to change this, such as community meetups, blog posts and position papers, but reviews of individual papers are not a suitable place.
> - We emphasise that LILO does NOT prioritise variability in responses or mistakes, but rather variability in rewards. It does not matter whether the model makes the same mistake every time, or a different mistake every time. If all samples are incorrect, it is a mathematical fact that existing policy gradient algorithms will not learn, no matter how many updates.
> - If there are some correct and some incorrect examples, and thus some gradient signal, it is an interesting hypothesis to consider that it might be easier to learn to fix 1 mistake than 10 different ones, and we thank the reviewer for this suggestion for future work.
>
>
> **Impact of theorem 3.1:**
> - We collected over ~10,000 gradient samples from the checkpoints used during training on GSM8K and MATH and found a statistically significant correlation between learnability and $ || \nabla_\theta J(\theta)||^2$, as show in the table below. See the rebuttal to reviewer GyS9 for more information on this.
>
> |                     | **GSM8K** |         | **MATH** |         |
> |---------------------|-----------|---------|----------|---------|
> | Learning algorithm  | **Corr**  | **Sig** | **Corr** | **Sig** |
> | REINFORCE           | 0.1629    | Yes     | 0.3145   | Yes     |
> | RLOO                | 0.1485    | Yes     | 0.4262   | Yes     |
> | GRPO (normalised)   | 0.1293    | No      | 0.3842   | Yes     |
> | GRPO (unnormalised) | 0.1690    | Yes     | 0.4178   | Yes     |
> | PPO                 | 0.3786    | Yes     | 0.7140   | Yes     |
> | VinePPO             | 0.3786    | Yes     | 0.7140   | Yes     |
>
> This table shows the correlation between learnability and $|| \nabla_\theta J(\theta) ||^2 $.
>
> - The theorem is the first to explain this phenomena and directly led to our method, which in turn led to significant improvement and acceleration of LLM training methods. Given the popularity of LLM reasoning, this constitutes significant downstream impact of Theorem 3.1.
> - Looking at the update magnitudes produced during training, we see that selecting the highest learnability produces 1.46x larger updates than not using learnability. Theorem 3.1 explains this, but it also explains why prioritising learnability produces 1.44x larger updates than simply rejecting any questions that are always answered correctly or incorrectly, and 1.40x larger updates than prioritising the most difficult questions. Existing theory cannot explain this.
>
> **“Combining theoretical RL and LLM reasoning is strange”:**
> - The reviewer is correct to note that it has become commonplace in LLM papers to present methods or heuristics that work, without any attempt to explain or motivate them with theory.
> - It is our position that ideally new methods are presented with insightful theory that motivates and explains them. The required theory to explain our method did not previously exist, and thus we presented it alongside the method. We are not opposed to publishing a more general theory paper in future.
>
> **Wall clock time:**
> Whilst we did provide wall-time impact in Table 1, we agree that it would be more useful to the reader if speed-ups were reported in wall-clock time alongside speed-ups in training steps. We have therefore amended Table 2 to be the following:
>
> | Algorithm | Train Dataset | Speed-up (Steps) | Speed-up (Wall Time) | Final test accuracy (%) |
> |-----------|---------------|------------------|----------------------|-------------------------|
> | PPO       | MATH          | 2.5×             | 2.0x                 | 19.1 → 21.8             |
> | PPO       | GSM8K         | 1.9×             | 1.5x                 | 51.1 → 53.2             |
> | VinePPO   | MATH          | 3.2×             | 3.1x                 | 22.8 → 24.9             |
> | VinePPO   | GSM8K         | 3.3×             | 3.2x                 | 53.2 → 55.9             |
> | GRPO      | ORZ57K        | 1.6×             | 1.3x                 | 35.5 → 37.1             |
>
> We cannot upload graphs during this rebuttal, but commit to changing the x-axis on training curves, Figure 1, 2, 3 and 4 for future versions.
>
> **Learning stagnation:**
> - LILO does prioritise some trajectories over others, which is precisely why it speeds up training.
> - Crucially however, it does not ignore them forever: If the model has no insight into a certain problem, and is mostly wrong when answering it, it will have low learnability, and training on it would have lower impact than training on a higher learnability question. If no questions with higher learnability exist, LILO will still train on this question. Methods that don't use LILO gain no benefit from training on these questions earlier on in the process. The same is true for questions that model consistently gets right.
>
> We kindly request that the reviewer examines our further clarifications and assesses whether they satisfactorily address the concerns previously raised, potentially altering the evaluation of our work. We welcome any additional insights or comments that may be deemed pertinent.

---

> > ### Author Response · Authors · 2025-08-06
> >
> > Thank you again for your review. We've prepared a response to address the points that you raised. We would appreciate hearing your thoughts to begin a discussion about your comments. Note that your review had some overlap with reviewer GyS9, who increased their score in light of our response.

---

> > > ### Comment · Area_Chair_mgbq · 2025-08-06
> > > **Please engage**
> > >
> > > Dear reviewer, this is a kind reminder to please engage with the authors ASAP. The discussion period ends on August 8th, but please don't wait until the last moment.
> > >
> > > Best,
> > > Your AC

---

> > ### Comment · Reviewer_Qkcn · 2025-08-07
> >
> > I thank the authors for clarifying the concept of learnability and providing additional results on wall clock time. I will raise my score to weak accept.

---

### Official Review · Reviewer_xAAo · 2025-07-02

**Clarity:** 3
**Significance:** 2
**Originality:** 2
**Rating:** 4
**Confidence:** 4

**Summary:**

The paper discusses a key inefficiency in RL methods for post-training in LLMs - especially when the tasks are about some form of reasoning. The central observation is that problems are either consistently solved or are tooo hard such that the LLMs might consistently fail – perhaps because the LLMs are not expressive enough. Training/exploring over these problems provides near zero learning signal for policy gradient based RL algorithms – this is the central claim. The proposal is an algorithm that exploits a concept of “learnability” and then the paper presents a method based on the concept of learnability, an algorithm for sampling “learning-*optimal*” samples.

**Questions:**

With regards to the proof – don’t we need fairly strong assumptions that log policy gradients are independent of the advantages? would that hold in practice – if so how likely? Wouldn’t there be fairly important implications when this assumption is violated?

I put it in the weakness bit – I am not sure I understood why GRPO works even if it loses its dependability on learnability?

Also, what happens if there are potential issues in samples – GSM8K for e.g., has incorrect Q, A pairs. In some cases spurious correlations can help in answering (matching the gold answer … which are inherently incorrect) such Qs (or samples).


There is some dangling issue — line 121 – multiple attempts to replicate Deepseek R1-Zero [] …. ?

There is a claim that’s repeatedly made … – 3x reduction in training steps — shouldn’t that be “upto” … in select settings? Also, not sure I understand the non-congruency in experiments - PPO vs VinePPO are evaluated on similar setups, but GRPO on a completely different one?

**Ethical Concerns:**

["NO or VERY MINOR ethics concerns only"]

**Final Justification:**

I happy with their continued engagement and effort that they are putting for the review.

**Limitations:**

yes

**Quality:**

3

**Strengths And Weaknesses:**

There are a few things that I am concerned about and mostly these are to do with claims that are a tad too generous. For instance, Learnability is a tad too strong as a term, the presented approach is oriented at making the model efficiently learn to maximise its *score* on problems that provide a variable reward signal. It does not—and cannot—guarantee that the model is learning the correct reasoning process (can it?). A problem with high "learnability" (e.g., a 50% success rate ) is simply a problem where the model's current strategy is unstable. It's at the "frontier" of what the model can solve with its existing, potentially flawed, heuristics. My worry is focusing on these problems accelerates the process of either solidifying a flawed-but-successful strategy or discovering a new one. It optimizes for finding some way to get the right final answer. The way I see it is that the method is about the efficiency of the RL training loop, not the semantic correctness of the knowledge being acquired. Indeed, a more accurate, albeit less flashy name is something along the lines of “training signal variance” or perhaps even “algorithmic utility” – is this sample at a sweet spot of difficulty where it can effectively drive *parameter updates* – instead of “learnability”. From a cog sci/pedagogical perspective, I am not sure learnability really captures the core aspects. When we talk about learning, we intuitively mean acquiring the correct, generalizable, step-by-step process to solve a problem. This implies understanding, not just successful pattern matching. Surely, Theorem 3.1 cannot capture the generality of this.

Another, thing that was a tad overstated was a “Gap in the literature” – for instance, the draft itself recognizes previous work, such as Tzannetos et al. 2024, Havrilla et al., 2024, or Smaug – pal et al., 2024 to name a few have all touched upon, with observations and conjectures. Granted that this work takes a fairly known theoretical result & empirical observations from a simplified context and generalizes it. While this generalization is a valid contribution, framing it as closing a vital gap and providing a novel proof that maximises the expected improvement, to me, comes across as an overstatement.

I see that for GRPO, which normalizes the advantage, the expected gradient magnitude loses its dependency on learnability. Oddly the experiments indicate that LILO still works for GRPO … How can we reconcile this?


On the generalization gap between training and test accuracy – training accuracy can approach 95% while test accuracy stalls below 60%. This reinforces my earlier bit that the model isn't *learning* robust reasoning, it's just getting better at solving the specific problems it's trained on.


Further, I would have loved to see error analysis – what changed? Are there types of samples that have changed? Could you have solved it with say rejection sampling? Finding a causal link at inference time would have made it stronger. I can see the curriculum bit, that’s interesting, but perhaps understanding what happened at test time would be useful.

The paper as I see it seems to be a nice engineering heuristic, I do like the heuristic. The theorem, as it stands in the current draft, feels like reverse engineering to justify the methodology.

---

> ### Author Rebuttal · Authors · 2025-07-31
>
> We thank the reviewer for their time and valuable feedback that has helped us improve this work. We are encouraged by the positive comments around the effectiveness of our method for boosting performance of algorithms that train LLMs to reason with RL. We address the weaknesses and questions one-by-one below and hope this resolves any of the outstanding concerns and that the reviewer will consider revising their rating.
>
> **The naming of “learnability”:**
> - You’re completely correct that training signal variance, algorithmic utility (or even just reward variance) are great names for this quantity. However, there is a body of previous work [16,17,18] that terms this ‘learnability’ and as such, it makes sense to use this term.
>
> **"The model isn’t learning robust reasoning - it is just getting better at solving specific problems:"**
> - This is indeed very much a concern with LLM reasoning, RL or even just machine learning in general. It doesn’t feel like a particularly fair critique of this paper specifically, which does not claim to make improvements in this area.
> - Methods for improving the generalisation of LLM reasoning are definitely needed (we also suggest this in Section 7), and we thank the author for acknowledging this.
> - This is a method to speed up training, and boost the overall test accuracy achieved. Given the popularity and resource consumption of LLM reasoning, this is an equally worthwhile and impactful direction orthogonal to improving generalisation.
> - As with the term “learnability”, one can take issue with established phrases such as “learning to reason”. Given the prevalence of “learning to reason” RLVR papers, it is reasonable to use this term to avoid needlessly adding new language to this topic, but we also recognise its limitations.
>
> **What happens if GSM8K has incorrect Q,A pairs?**
> - As with any most machine learning methods, if the data is wrong or low quality, the model training would likely suffer.
> - There is an argument that LILO could help to reduce the negative effects of mislabelled data. For example, if the answer is wrong and the model consistently answers correctly but is deemed incorrect, the learnability of this sample would be 0, and we would ignore training on it. This is in contrast with not using LILO which would continually prioritise training on this question.
>
> **Theorem 3.1’s closing of a “vital gap”:**
> - There is a rich history of curricula for traditional RL settings, and as you note, Learnability has been proposed before, such as in [16].
> - We emphasise that the relevance of [16] on reasoning with LLMs is extremely small - there is no clear method for computing and prioritising learnability, there is no suggestion that this should work beyond the REINFORCE algorithm, and is it considered in traditional RL tasks.
> -  Beyond [16], it is non-trivial to decide which curricula methods from traditional RL are suited for the LLM reasoning setting:for example Havrilla et al tried some and these failed. This is true for other areas of RL too, for example value networks, which are popular in deep RL, have been shown to not transfer to the LLM setting.
> - This paper therefore provides the theory to explain why learnability based curricula are particularly well suited for the LLM setting. Previous work does not provide any theory as to why learnability would be useful for modern RL algorithms such as RLOO, GRPO, VinePPO and PPO.
> - Looking at the update magnitudes produced during training, we see that selecting the highest learnability produces 1.46x larger updates than not using learnability. Theorem 3.1 explains this, but it also explains why prioritising learnability produces 1.44x larger updates than simply rejecting any questions that are always answered correctly or incorrectly, and 1.40x larger updates than prioritising the most difficult questions. Existing theory cannot explain this.
> - Crucially, this results in LILO being the first to show that a curriculum method can boost overall performance on LLM reasoning tasks. Given the popularity and resource consumption of this setting this is highly impactful and useful to the community.
>
> **Assumptions within Theorem 3.1:**
> - We refer the reviewer to our rebuttal to reviewer GyS9, where we comprehensively address this with statistical testing of our assumptions and the theory.
>
> **LILO + GRPO:**
> - Thank you for pointing this out! We definitely should have made this clearer. We use the unnormalised version of GRPO that has shown to outperform normalised GRPO for reasoning tasks [cite DrGRPO and Replicating R1]. As we point out in Section 3, using the unnormalised GRPO does benefit from prioritising learnability.
> - We wanted to ensure that the experimental setups we were using already had optimised baselines, and therefore that LILO’s improvement could not be brought about e.g by tuning hyperparameters. We therefore used the VinePPO library which provides optimised baselines of PPO and VinePPO for GSM8K and MATH, and the Oat library which provides optimised baselines of GRPO on ORZ57K.
>
> **Error analysis and rejection sampling:**
> - In Figure 5 we visualise how the number of reasoning steps chosen by LILO changes throughout training. This is strong evidence that rejection sampling would not have been sufficient - questions with a higher number of reasoning steps only become learnable later on during training.
> - We agree that further visualisation here of the exact questions that progressed from hard to easy throughout training would be useful to the reader, and we commit to adding these in the camera-ready version.
>
> **Errata:**
> - Thank you for your diligence in picking up on the mistake on line 121, we have fixed this.
> - Thank you for the suggestion of rewording to use “up to 3x reduction”. This is indeed a fairer reflection of Table 2.
>
> We respectfully ask the reviewer to evaluate our added details: we believe we comprehensively address any concerns, and that as such it would be reasonable to modify the assessment of our contribution. We are open to any additional feedback or relevant points you might wish to discuss.

---

> > ### Author Response · Authors · 2025-08-06
> >
> > Thank you again for your review. We've prepared a response to address the points that you raised. We would appreciate hearing your thoughts to begin a discussion about your comments.

---

> > > ### Comment · Area_Chair_mgbq · 2025-08-06
> > > **Please engage**
> > >
> > > Dear reviewer xAAo, this is a kind reminder to please engage with the authors ASAP. The discussion period ends on August 8th, but please don't wait until the last moment.
> > >
> > > Best,
> > > Your AC

---

> > ### Comment · Reviewer_xAAo · 2025-08-07
> >
> > Thank you for the response and for engaging with the feedback provided. I appreciate the clarifications and the commitment to make changes in the final version. However, several of my core concerns remain.
> >
> > -- On the term 'Learnability': I appreciate the desire for consistency with prior literature. However, I must respectfully disagree with the justification provided. The cited works [16, 17, 18] are very clearly situated within the context of traditional reinforcement learning tasks, as is evident from their titles, abstracts, and content. In that domain, 'learnability' has a certain specified meaning. My concern is that applying this same term to the complex domain of LLM reasoning is misleading. When we usually discuss 'learning to reason', there is a strong, implicit connotation of acquiring a correct and generalizable cognitive process, not just optimizing pattern matching to maximize a score. Not sure I agree with the rationale here.
> >
> > -- On Robust Reasoning vs. Solving Specific Problems bit: The response seems to double down on the position that this is a general problem in machine learning. While I agree that generalization is a universal challenge, my critique was directed specifically at the claims and interpretation within this paper. The significant gap you report between training accuracy (approaching 95%) and test accuracy (stalling below 60%) is a key result of your own work. This strongly suggests that the model is not learning robust reasoning but is instead becoming highly adept at solving the specific problems it is trained on. Your method, LILO, accelerates this process. Therefore, the core of my point remains: isn't the paper is primarily about improving the efficiency of the RL training loop, not necessarily the semantic correctness or generalizability of the knowledge being acquired. I feel like this distinction is crucial and should be reflected more clearly in the paper's claims.
> >
> > -- On Theorem 3.1 and the Vital Gap bit: I acknowledge that generalizing a known theoretical result to modern RL algorithms in the LLM context is indeed a valid and a good contribution. However, I maintain that framing this as closing a 'vital gap' in the literature seemingly is an overstatement.
> >
> > -- On Experimental Incongruence and Lack of Error Analysis: I do not believe my question about the incongruent experimental setups was fully addressed. The rationale given was that you used pre-optimized baselines from different libraries. However, this choice makes it very difficult to fairly assess the impact of LILO. by evaluating PPO/VinePPO on MATH/GSM8K and GRPO on a different dataset (ORZ57K), you introduce confounding variables that make it impossible to compare the behaviour of *+LILO across different base algorithms. A more rigorous experimental design would have involved comparing these algorithms on the same tasks to isolate the effects of your proposed method.
> >
> > This ties directly into my other point about error analysis. For instance, you show a modest but interesting performance boost for VinePPO+LILO on MATH (from ~22% to ~24%). It would be incredibly insightful to understand what changed.
> >
> > What kinds of problems was the model now able to solve?
> >
> > Where did this ~2% performance gain come from?
> >
> > Are the types of problems that saw improvement with PPO+LILO on MATH similar to the types of problems that saw improvement with GRPO+LILO on ORZ57K, or are they different?
> >
> > Without this deeper analysis, to me, it is difficult to understand what is actually being 'learned' and to substantiate the claims beyond LILO being an effective engineering heuristic for accelerating training.

---

> ### Author Response · Authors · 2025-08-08
>
> Dear Reviewer xAAo,
>
> Thanks so much for the continued discussion, we have considered your response and prepared a reply to each point below (note we had to spread this over 3 comments)
>
> **On the terms ‘learnability’ and "learning to reason"**
> - We completely agree that “learning to reason” should be reserved for the discussion of learning correct and generalisable cognitive processes. Sadly, it is our experience that this isn’t reflected in much of the recent literature. It seems to us that “Learning to reason” has become a popular term for improving LLMs' ability at “reasoning style” questions (maths, science, logic etc),  often using reinforcement learning. In additional to the seminal OpenAI O1 release [1], here are 4 papers from 2025 that all have “learning to reason” in the title [2,3,4,5], and 6 papers that all refer to “learning” and “reasoning” [6,7,8,9,10,11].  All of these papers claim that the model has “learnt to reason” via evaluation on holdout problems, and as far as we can tell do not obviously do any deep analysis on the type or generalisability of the cognitive processes used. If one is happy to use “learning to reason” to mean improving test accuracy, it seems reasonable to use “learnability” to mean questions that when trained on yield a higher test accuracy.
> - On the other hand, you are correct that there is an alternative strand of work that pushes back against this, with papers such as [12,13,14,15,16] that analyse whether the “reasoning” learnt using RL on LLMs is actually a robust, generalisable, cognitive process.  Since our work is directly aimed at those practising RL on LLMs for maths style tasks, we chose to go with the former “learning to reason” terminology, but we appreciate that it can be misleading for those more familiar with the latter referenced work. We’ve suggested how we might perhaps find a compromise between these two styles below, but are open to your suggestions.
> - You’re right to point out that previous work on learnability has been only in “traditional RL tasks”. We actually have a similar gripe to the reviewer when ‘learnability’ is used in traditional settings. When teaching an agent to solve mazes: is selecting *learnable* training levels ones where it might learn general principles for solving mazes (ie the right hand rule), or are these just levels that haven’t yet been memorised? However, learnability has become the standard term for reward variance in traditional settings. Since Theorem 3.1 applies to any sparse MDP, whether in the traditional RL or LLM setting, we chose to stick with the term, instead of redefining it, but we can appreciate how for some readers this may come with added connotations. We're happy to add an addendum/clarifying note to this effect, if you see fit (see below for suggestions).
> - This leaves us in a sticky spot - concept drift like this is annoying, especially when some (often closely related) areas of ML overload the same term. We’ve done our best to balance concept drift with situating work within existing literature and terminology which is also important for clarity, for paper visibility, and for giving due credit to the lineage of ideas.
> - Honestly, we would greatly appreciate the reviewer's guidance on how best to resolve this - would they be happy with some reconciling / defining terms early on in the paper, such as:
>     - By learning to reason, we specifically mean here “learning to solve unseen maths problems through training on a set of existing problems”. By “learnable problems”, we simply mean problems with high variance in rewards for which the model will likely obtain high gradient updates. `
>
>     Also perhaps in limitations, we could be more explicit:
>
>     - Our method helps improve the overall accuracy achieved on the unseen test set, and does so in fewer training steps. Our method does not directly aim to solve the broader goal of ensuring the model is learning correct and generalisable cognitive processes.
>
>     We hope this would suffice, but if not, we could consider changing the name of the paper. It would be a shame for an issue of semantics to interfere with a paper whose broader contribution we really believe in.
>     Please do let us do know your opinions on how best to resolve this dilemma.

---

> > ### Author Response · Authors · 2025-08-08
> >
> > **On ‘robust reasoning vs solving specific problems’:**
> > - The above section covers a lot of this, and we agree that there isn’t a clear distinction in this paper between “learning robust reasoning” and just improving on test problems. This has become standard recently due to the liberal use and weakening of terms like “learning to reason”. We hope that adding the above definitions and limitations would clarify this for any readers, but let us know any suggestions for how else we might make this clearer.
> > - Perhaps one source of misunderstanding between the reviewer and ourselves is that we believe we make very clear in the paper that LILO does not aim to improve generalisation or robustness of reasoning. In Appendix E Figure 10 we examine the ratio between train and test scores and find it unchanged for LILO. It seems unfair to us to therefore label the generalisation gap “a key result of your own work”. This is further stated in Section 7 with the subsection titled “Learnability does not solve RL’s generalisation gap”. Please do let us know if we could make this clearer in any way.
> > - We do claim that LILO increases accuracy at unseen maths problems similar to those trained on (Figures 1, 2 and 3), and does so in fewer steps. We also show modest improvements in accuracy at unseen maths problems of different styles and difficulties from other hold out datasets (Appendix E Table 3). We believe that the stating and evidence for this claim is clearly and thoroughly laid out, but welcome any suggestions for how we might make this more so.
> >
> > **On Theorem 3.1 being termed a ‘vital gap’**
> > - We think reviewer’s criticism of this is reasonable and thank them for acknowledging our contribution. This may already be clear to the reviewer but for the avoidance of any doubt, we must emphasise that prior work across both LLM and “traditional RL” is limited to a) sparse deterministic binary MDPs b) a loose consideration of the case where all rewards are 0s or all are 1s, making learnability = 0 c) a loose non mathematical argument for 0 ≤ learnability ≤ 1 for the REINFORCE algorithm. Theorem 3.1 is the first concrete mathematical argument for a) any sparse MDP (not necessarily deterministic or binary) b) 0 ≤ learnability ≤ 1 and c) for modern algorithms such as RLOO, PPO, VinePPO and GRPO. It is applicable in both the LLM and ‘traditional’ settings to explain why learnability maximisation works.
> > - However we are not strongly wedded to using the term “vital”, so would commit to using perhaps “fills a gap in the literature” in the camera ready version?
> >
> > **On further experiments and analysis**
> > - We thank the reviewer for these suggestions. We hope that we have presented sufficient evidence for our central claim that LILO is a mathematically justified way to enable existing highly tuned methods to reach higher accuracies on unseen test problems, and to do so in fewer training steps.
> > - Of course we do agree that additional experiments would allow for the comparison of LILO’s impact across existing algorithms, which would be an interesting avenue for future exploration. The additional error analysis would also be interesting and would make the paper stronger. We are doing our utmost  to present both of these for the reviewer before the end of the rebuttal window.
> >
> > In summary, thank you again for these excellent points. We are in agreement about the difficulty of maintaining consistency with prior literature whilst also preventing the dilution of phrases like “learnability” or “learning to reason”. Hopefully we have supplied you with some agreeable ways forward, but would greatly welcome advice about how you would address this challenge if you were in our position.
> >
> > Much of the rest of our discussion has focussed on helping us be very clear in our central claim that “LILO is a mathematically justified way to enable existing highly tuned methods to reach higher accuracies on unseen test problems, and to do so in fewer training steps”. We hope that we have clarified our position both on LILO’s impact on generalisation, and on how additional results and experiments would be interesting (and we're working around the clock to get them before the end of the window) but are not themselves necessary for proving our central claim.
> >
> > Please do let us know of any additional feedback or points you may wish to discuss.

---

> ### Author Response · Authors · 2025-08-08
>
> [1] **OpenAI** (2024) *Learning to reason with language models*. Available at: https://openai.com/index/learning-to-reason-with-llms/ (Accessed: 8 August 2025)
>
> [2] Chen, M., Li, T., Sun, H., Zhou, Y., Zhu, C., Wang, H., Pan, J.Z., Zhang, W., Chen, H., Yang, F. and Zhou, Z., 2025. Learning to reason with search for llms via reinforcement learning. *arXiv preprint arXiv:2503.19470*.
>
> [3] Zhao, X., Kang, Z., Feng, A., Levine, S. and Song, D., 2025. Learning to reason without external rewards. *arXiv preprint arXiv:2505.19590*.
>
> [4] Yan, J., Li, Y., Hu, Z., Wang, Z., Cui, G., Qu, X., Cheng, Y. and Zhang, Y., 2025. Learning to reason under off-policy guidance. *arXiv preprint arXiv:2504.14945*.
>
> [5] Zhang, J., Huang, J., Yao, H., Liu, S., Zhang, X., Lu, S. and Tao, D., 2025. R1-vl: Learning to reason with multimodal large language models via step-wise group relative policy optimization. *arXiv preprint arXiv:2503.12937*.
>
> [6] Dang, Q.A. and Ngo, C., 2025. Reinforcement Learning for Reasoning in Small LLMs: What Works and What Doesn't. *arXiv preprint arXiv:2503.16219*.
>
> [7] Team, K., Du, A., Gao, B., Xing, B., Jiang, C., Chen, C., Li, C., Xiao, C., Du, C., Liao, C. and Tang, C., 2025. Kimi k1. 5: Scaling reinforcement learning with llms. *arXiv preprint arXiv:2501.12599*.
>
> [8] Havrilla, A., Du, Y., Raparthy, S.C., Nalmpantis, C., Dwivedi-Yu, J., Zhuravinskyi, M., Hambro, E., Sukhbaatar, S. and Raileanu, R., 2024. Teaching large language models to reason with reinforcement learning. *arXiv preprint arXiv:2403.04642*.
>
> [9] Guo, D., Yang, D., Zhang, H., Song, J., Zhang, R., Xu, R., Zhu, Q., Ma, S., Wang, P., Bi, X. and Zhang, X., 2025. Deepseek-r1: Incentivizing reasoning capability in llms via reinforcement learning. *arXiv preprint arXiv:2501.12948*.
>
> [10] Wang, Y., Yang, Q., Zeng, Z., Ren, L., Liu, L., Peng, B., Cheng, H., He, X., Wang, K., Gao, J. and Chen, W., 2025. Reinforcement learning for reasoning in large language models with one training example. *arXiv preprint arXiv:2504.20571*.
>
> [11] Rastogi, A., Jiang, A.Q., Lo, A., Berrada, G., Lample, G., Rute, J., Barmentlo, J., Yadav, K., Khandelwal, K., Chandu, K.R. and Blier, L., 2025. Magistral. *arXiv preprint arXiv:2506.10910*.
>
> [12] Hazra, R., Venturato, G., Martires, P.Z.D. and De Raedt, L., 2025. Have Large Language Models Learned to Reason? A Characterization via 3-SAT Phase Transition. *arXiv preprint arXiv:2504.03930*.
>
> [13] Li, Z.Z., Zhang, D., Zhang, M.L., Zhang, J., Liu, Z., Yao, Y., Xu, H., Zheng, J., Wang, P.J., Chen, X. and Zhang, Y., 2025. From system 1 to system 2: A survey of reasoning large language models. *arXiv preprint arXiv:2502.17419*.
>
> [14] Shojaee, P., Mirzadeh, I., Alizadeh, K., Horton, M., Bengio, S. and Farajtabar, M., 2025. The illusion of thinking: Understanding the strengths and limitations of reasoning models via the lens of problem complexity. *arXiv preprint arXiv:2506.06941*.
>
> [15] Zhang, Y., Wang, H., Feng, S., Tan, Z., Han, X., He, T. and Tsvetkov, Y., 2024. Can LLM Graph Reasoning Generalize beyond Pattern Memorization?. *arXiv preprint arXiv:2406.15992*.
>
> [16] Xu, F., Lin, Q., Han, J., Zhao, T., Liu, J. and Cambria, E., 2025. Are large language models really good logical reasoners? a comprehensive evaluation and beyond. *IEEE Transactions on Knowledge and Data Engineering*.

---

> > ### Author Response · Authors · 2025-08-09
> >
> > Dear reviewer xAAo,
> >
> > We are delighted to now be able to provide you with some of the further experiments and analysis you suggested.
> >
> > **Results for GRPO+-LILO on MATH & GSM8K**
> >
> > Thank you for suggesting this experiment. Whilst, we maintain that it is not strictly necessary as evidence of the central claim of this paper, it is nice to be able to directly compare the effect of LILO on different algorithms, and has strengthened the paper.
> >
> > After implementing GRPO in the VinePPO library to enable direct comparison, we were able to run 120 training steps before the end of the rebuttal window yielding:
> >
> > | Method          | GSM8K | MATH |
> > |-----------------|-------|------|
> > | PPO:            | 43.8  | 16.8 |
> > | PPO + LILO:     | 47.4  | 18.0 |
> > | VinePPO:        | 48.9  | 19.7 |
> > | VinePPO + LILO: | 52.0  | 20.7 |
> > | GRPO:           | 44.7  | 20.0 |
> > | GRPO + LILO:    | 46.9  | 18.1 |
> >
> > We would commit to putting the full results for 1000 training steps in the paper.
> >
> > **Question analysis**
> >
> > Thanks for this suggestion - We uniformly sampled 100 questions from the MATH dataset and evaluated PPO and PPO+LILO on them.
> >
> > We observed that:
> >
> > Both Correct: 11 questions
> >
> > Only PPO Correct: 11 questions
> >
> > Only PPO + LILO Correct: 14 questions
> >
> > Neither Correct: 64 questions
> >
> > **note this suggests a test accuracy of 22% for PPO and 25% for PPO+LILO. These new values were computed only over 100 questions to obtain examples of answers for each model. They are not statistically significant evidence (p values of 0.33 and 0.44 respectively) for rejecting the null hypothesis of the values reported in the paper (19.1% and 21.8%), which were computed over 1000+ problems.*
> >
> > Inspecting the answers to these questions for some time, we could not find any obvious differences in patterns of behaviour between the two models.
> >
> > As a sanity check, we put the entire results file into Gemini Pro, with the following prompt:
> > *”These are the questions and responses from two models. The four files are questions that both got correct, model 1 got correct but model 2 got wrong, etc. Please look through the responses and see if there are any patterns in the behaviour differences between the two models.”*
> >
> > It answered:
> >
> > - The PPO model exhibits a profound lack of conceptual understanding across various topics. It often misapplies or invents formulas, leading to incorrect solutions. Its errors are frequent, varied, and often basic. They range from simple arithmetic mistakes to major conceptual failures, making it untrustworthy.
> > - The PPO+LILO model generally applies the correct mathematical concepts and formulas. It successfully solves problems requiring knowledge of combinations, modular arithmetic, the Least Common Multiple, and the quadratic discriminant. Its errors are more often on highly complex problems rather than a misunderstanding of core principles.
> > On a few occasions, it entered a loop and produced repetitive, nonsensical text instead of a valid solution. However, when it avoids this bug, its mathematical reasoning is typically sound.
> >
> > We could see some evidence that this is a reasonable conclusion, but of course as with any LLM produced analysis, it must be taken with a pinch of salt.

---

### Official Review · Reviewer_GyS9 · 2025-07-03

**Clarity:** 3
**Significance:** 3
**Originality:** 2
**Rating:** 4
**Confidence:** 4

**Summary:**

Learning from questions may be inefficient as the levels of difficulty of the questions do not align with the model being trained. This paper proposes to prioritize training on questions with high variance of success referred to as learnability. A theory is provided to prove that the proposed method, termed LILO, maximizes the expected improvement of the model. Experiments validate the effectiveness of LILO and show that LILO can greatly reduce the number of training steps.

**Questions:**

-	In Theorem 3.1, why $g_{t,\theta}$ is assumed to be independent of $A_{t’}$?
-	Why does the argument that expected policy improvement increases with the variance of the final reward hold, given that the expected value of the sum of $g_{t,\theta}$ could be decreased?
-	Have your explored other sample selection methods?

**Ethical Concerns:**

["NO or VERY MINOR ethics concerns only"]

**Final Justification:**

The authors’ response has addressed my main concerns regarding the originality of the paper and the validity of Theorem 3.1. I have increased my score accordingly.

**Limitations:**

yes

**Quality:**

2

**Strengths And Weaknesses:**

**Strengths:**
- The proposed method is very simple and can be added seamlessly to existing training libraries with minimal changes.
- The paper is generally well-written and easy to follow.
- Experimental results are promising. The proposed sample selection method greatly speeds up the training as well as improves model performance.

**Weaknesses:**

*Originality*:

-	The proposed method in a simple variation of hard sampling mining in the RL setting. Based on the definition of learnability, which is the variance of binary outcomes of a question, it is maximized when the success rate is 0.5. In this case, the model is completely uncertain about the question with equal rates of success and failure, and the selected question is most challenging to the model.

*Quality*:

-	Some important assumptions are not well justified in the paper. Specifically, in Theorem 3.1, $g_{t,\theta}$ is assumed to be independent of $A_{t’}$. This needs more justification.
-	Theorem 3.1 only states that the expected increment in reward depends on the variance of final reward. Maximizing the variance does not necessarily indicate that the expected reward will also be increased. Moreover, based on the independence assumption, the expected value of the sum of $g_{t,\theta}$ may be decreased. Therefore, the claim that learnability optimally improves LLMs is overstated.

*Typos*:

-	Line 87: asses -> assess
-	Line 102: J(\theta_k ->  J(\theta_k)
-	Line 121: empty citation

---

> ### Author Rebuttal · Authors · 2025-07-31
>
> We thank the reviewer for their time and valuable feedback that has helped us improve this work. We are encouraged by the positive comments around the both simplicity and effectiveness of our approach at boosting the performance of RLVR algorithms.  We address the weaknesses and questions one-by-one below and hope this resolves any of the outstanding concerns and that the reviewer will consider revising their rating.
>
> **Originality:**
> - You are completely right to note the long history of studying hard sample mining in the traditional RL setting. We were inspired by many of these works especially Prioritised Level Replay [26] to try to produce a simple and effective method for the LLM RLVR domain. (For the avoidance of confusion, we note that our samples are not the hardest samples, they are the ones at the correct level of difficulty, thus providing the largest gradient update)
> - Beyond hard sample mining, there is a rich history of many other types of curricula for traditional RL settings, and as you note, Learnability has been proposed before, such as in [16].
> - We emphasise that the relevance of [16] on reasoning with LLMs is extremely small - there is no clear method for computing and prioritising learnability, there is no suggestion that this should work beyond the REINFORCE algorithm, and is it considered in traditional RL tasks.
> -  Beyond [16], it is non-trivial to decide which curricula methods from traditional RL are suited for the LLM reasoning setting:for example Havrilla et al tried some and these failed. This is true for other areas of RL too, for example value networks, which are popular in deep RL, have been shown to not transfer to the LLM setting.
> - We identified that Learnability based curricula are particularly well suited for the LLM setting because it is non-stochastic, and uses sparse, binary rewards. Even once we identified this, there was still work to do to adapt it to the LLM setting. For example, in Appendix  E Figure 11 we detail how the frequency with which you recalculate learnable questions, which is typically done every 10-50 timesteps in the traditional setting, is catastrophic in the LLM setting.
> - Crucially, this results in LILO being the first to show that a curriculum method can boost overall performance on LLM reasoning tasks. Given the popularity and resource consumption of this setting this is highly impactful and useful to the community.
>
> **Quality (The validity of Theorem 3.1):**
>
> - We greatly appreciate the valuable remarks regarding Theorem 3.1. Prompted by the reviewer’s comments, we have revisited the proof of Theorem 3.1, and realized that the assumptions needed for the proof can be relaxed. Previously we assumed independence between $g_{t, \theta}$ and $A_{t’}$, for any $t$, $t’$. The new, less restrictive, assumptions we need, are as follows:
>     - For REINFORCE and RLOO we require that $|| \sum_{t=0}^{T-1} g_{t, \theta} ||^2$ is uncorrelated with $A_t = r(s_T)^2$ in the case of REINFORCE, and uncorrelated with $A_t = ( (r(s_T) - \mathbb{E}{\pi_\theta} [ r(s_T) ])^2$ in the case of RLOO. This is a less restrictive assumption than the independence between $A_t$ and $g_{t’, \theta}$, for any $t$, $t’$, and lack of correlation is much easier to test for than independence.
>     - For PPO and VinePPO we now only require that $A_t^2 = (V^{\pi_\theta}(s_{t+1}) - V^{\pi_\theta}(s_t))^2$ and $|| g_{t, \theta’} ||^2$ are uncorrelated for any $t = 0, …, T-1$. This is a less restrictive assumption due to again replacing independence with lack of correlation, and since we no longer need any assumption on the dependence of $A_t$ and $g_{t’, \theta}$, at different times $t \neq t’$.
> - We in particular appreciate the excellent comment that $|| g_{t, \theta} ||^2$ or $ || \sum_{t = 0}^{T-1} g_{t, \theta} ||^2$ could depend on learnability. We completely agree that those terms could depend on the underlying question of the MDP, and thus on the learnability of that question. We have thus also tested the correlation of those terms with learnability, in addition to all the revised assumptions made in the theorem. Below are all the details and results regarding those statistical tests:
> - For REINFORCE and RLOO:
>     - We randomly sampled questions, answers and gradient locations from the training process, drawing ~10,000 datapoints for each of the MATH and GSM8K experiments.
>     - We found a very small correlation between a) $|| g_{t, \theta} ||^2$ and $A_t$ b) $||g_{t, \theta}||^2$ and $t$, and c) $ || \sum_{t=0}^{T-1} g_{t, \theta} ||^2 $ and learnability. Testing end to end, we found that these small negative correlations are dwarfed by the increase of learnability itself. As explained by our theorem, $ || \nabla_\theta J(\theta) ||^2$ has a strong statistically significant positive correlation with learnability. These are shown in the tables below.
> - For PPO, VinePPO: As mentioned above, we have relaxed the assumptions needed for our proof, and now only require that:
>     - $ ||g_{t, \theta} ||^2$ and $(V^{\pi_\theta}(s_{t+1}) - V^{\pi_\theta}(s_t))^2$ are uncorrelated, for any $t = 0., …, T-1$. We found no statistically significant correlation between those terms, when testing this.
>     - $ \mathbb{E} \[ ||g_{t, \theta} ||^2 \]$ does not depend on $t$. We found no statistically significant correlation between $|| g_{t, \theta} ||^2$ and $t$ when testing this.
>     - $ ||g_{t, \theta} ||^2 $ does not depend on learnability. We found no statistically significant correlation between $ ||g_{t, \theta} ||^2 $ and learnability when testing this.
>     - Testing end to end, we found that as explained by the final result of our theorem, $|| \nabla_\theta J( \theta) ||^2$ has a strong statistically significant correlation with learnability.
> - To summarize, we have thoroughly tested empirically the validity of the assumptions in Theorem 3.1, and conclude that those are reasonable assumptions to make. We thus believe that Theorem 3.1 provides a well justified quantitative explanation of how LILO works under the hood.
>
> |                                                                                            | **GSM8K** |         | **MATH** |         |
> |--------------------------------------------------------------------------------------------|-----------|---------|----------|---------|
> | **Assumption**                                                                             | **Corr**  | **Sig** | **Corr** | **Sig** |
> | **REINFORCE/RLOO/GRPO (Appendix B, Equation 6)**                                           |           |         |          |         |
> | No correlation between $r(s_T)^2$ and $\|\|\sum_{t=0}^{T-1}g_t\|\|^2$                      | -0.0460   | Yes     | -0.0315  | Yes     |
> | No correlation between $(r(s_T)-\mathbb{E}[r(s_T)])^2$ and $\|\|\sum_{t=0}^{T-1}g_t\|\|^2$ | -0.0169   | Yes     | -0.0211  | Yes     |
> | No correlation between Learnability and $\|\|\sum_{t=0}^{T-1}g_t\|\|^2$                    | -0.0327   | Yes     | -0.0319  | Yes     |
> |                                                                                            |           |         |          |         |
> | **VinePPO / PPO (Tightened proof, above)**                                                 |           |         |          |         |
> | No correlation between $\|\|g_t\|\|^2$ and $V(s_{t+1}) - V(s_t)$                           | -0.0025   | No      | -0.0030  | No      |
> | No correlation between $\|\|g_t\|\|^2$ and $t$                                             | -0.0072   | No      | -0.0046  | No      |
> | No correlation between $\|\|g_t\|\|^2$ and learnability                                    | -0.0.0057 | No      | 0.0105   | No      |
>
>
> |                     | **GSM8K** |         | **MATH** |         |
> |---------------------|-----------|---------|----------|---------|
> | Learning algorithm  | **Corr**  | **Sig** | **Corr** | **Sig** |
> | REINFORCE           | 0.1629    | Yes     | 0.3145   | Yes     |
> | RLOO                | 0.1485    | Yes     | 0.4262   | Yes     |
> | GRPO (normalised)   | 0.1293    | No      | 0.3842   | Yes     |
> | GRPO (unnormalised) | 0.1690    | Yes     | 0.4178   | Yes     |
> | PPO                 | 0.3786    | Yes     | 0.7140   | Yes     |
> | VinePPO             | 0.3786    | Yes     | 0.7140   | Yes     |
>
> This table shows the correlation between learnability and $|| \nabla_\theta J(\theta) ||^2$.
>
> We invite the reviewer to consider our supplementary explanations and assess if they resolve the raised concerns, possibly influencing their appraisal of our work. Should there be any other aspects or feedback you find relevant, please do not hesitate to share.

---

> > ### Comment · Reviewer_GyS9 · 2025-08-04
> >
> > Thank you for your response, which has addressed my main concerns regarding the originality of the paper and the validity of Theorem 3.1. I have increased my score accordingly.

---

### Decision · Program_Chairs · 2025-09-17

**Decision:**

Accept (poster)

**Comment:**

The reviewers converged on the view that the paper makes a clear and well-substantiated contribution: LILO provides a simple, mathematically motivated sampling strategy that accelerates RL training for reasoning tasks with LLMs and yields consistent accuracy improvements across datasets. Multiple reviewers highlighted the clarity and writing quality, the strong empirical evidence, and the usefulness of the theoretical connection between reward variance and policy improvement. While some initially questioned the originality and assumptions of Theorem 3.1, the authors’ rebuttal and additional experiments addressed these concerns convincingly, leading reviewers to raise their scores.

The main disagreements centered on scope and framing. Reviewer xAAo argued that the term “learnability” (and by extension “learning to reason”) risks overstating the contribution, since LILO improves training efficiency rather than robust reasoning generalization, but in my view "learnability" is a well established term for reward variance. Others questioned novelty relative to earlier RL curricula work and suggested that claims about closing a “vital gap” were somewhat overstated, which I agree with. Nonetheless, even these critical reviewers accepted the value of the heuristic, acknowledged the theoretical generalization to modern RL algorithms as a meaningful contribution, and increased their evaluations after discussion. With four reviewers aligned that the central claim is supported by both theory and experiments, the overall consensus favors acceptance.